# Bispecific GLP-1/GLP-2 agonism in advanced type 2 diabetes: preclinical characterization and a randomized, double-blind, placebo-controlled phase I trial

PG-102 is a potency-optimized bispecific Fc fusion protein targeting GLP-1 and GLP-2 receptors. In *db/db* mouse models of advanced diabetes characterized by uncontrolled hyperglycemia and catabolic weight loss, PG-102 achieved superior and sustained glycemic control compared with semaglutide or tirzepatide while preserving body weight, uncoupling glycemic control from catabolic weight loss. Mechanistic studies indicated that these effects were not driven by acute insulinotropic activity, but by β-cell preservation and enhanced glucose uptake. These benefits required dual GLP-1R/GLP-2R engagement, as PG-102 outperformed monospecific Fc fusion agonists or their combination and promoted coordinated receptor trafficking with delayed internalization. We conducted a randomized, double-blind, placebo-controlled multiple ascending dose phase 1 study at a single center in the Republic of Korea in adults with overweight (BMI 25–30 kg/m²). Twenty-four participants were randomized within three weekly dose cohorts (15 mg, 30 mg, and 30/60 mg; $n = 6$ per cohort) in a 6:2 ratio to receive PG-102 ($n = 18$) or placebo ($n = 6$). All randomized participants (PG-102 $n = 18$; placebo $n = 6$) received at least one dose and were included in the safety analysis. Safety and tolerability were predefined as the primary endpoint and assessed by treatment-emergent adverse events (TEAEs). TEAEs occurred in 5/6 (83.3%) participants in each PG-102 cohort and 4/6 (66.7%) in placebo; treatment-related AEs occurred in 5/6 (83.3%) and 3/6 (50.0%), respectively. No serious adverse events or discontinuations due to adverse events occurred. Gastrointestinal events were mild to moderate. These findings support bispecific GLP-1/GLP-2 agonism as a mechanistically distinct incretin strategy in advanced T2D. ClinicalTrials.gov identifier: NCT06309667.

Glucagon-like peptide-1 receptor agonists (GLP-1RAs) are standard-of-care agents that provide potent glucose-lowering without hypoglycemia, together with weight reduction and cardiovascular and renal benefits. Building on this success, the field has progressed toward dual and triple agonists to further enhance metabolic outcomes, exemplified by GLP-1/GIP co-agonists that deliver superior glycemic and weight control[1]. Nevertheless, fewer than half of patients treated with GLP-1RAs achieve guideline-recommended glycemic targets (typically

✉e-mail: ycsung@postech.ac.kr

HbA1c < 7.0%), and treatment discontinuation remains a major challenge, driven largely by gastrointestinal side effects and inadequate glycemic control, with ~50% of patients discontinuing therapy within 12 months of initiation[2–4]. Moreover, current GLP-1RAs are not designed for patients experiencing unintentional weight loss—a frequent manifestation of advanced type 2 diabetes marked by catabolism and muscle wasting[5–7]. These gaps highlight the need for agents with improved tolerability that can sustain glycemic control without excessive weight loss, particularly in elderly, lean, or sarcopenic individuals for whom preservation of body mass is critical.

GLP-2 receptor agonism has emerged as an underutilized yet promising complement to GLP-1RA therapy, offering unique advantages for T2D treatment. Traditionally recognized for its role in intestinal growth and repair, GLP-2 has also demonstrated significant potential in metabolic regulation. Unlike GLP-1, GLP-2 does not substantially enhance acute insulin secretion, but instead supports β-cell health by promoting proliferation and protecting against cytokine-induced apoptosis under metabolic stress[8,9]. In addition, GLP-2 enhances insulin sensitivity and peripheral glucose disposal, as evidenced by improved glucose tolerance in vivo without concomitant rises in circulating insulin, as well as increased glucose uptake in human adipocytes[9,10]. GLP-2 also exerts anti-inflammatory actions in islets, including suppression of pro-inflammatory cytokines such as IL-1β, thereby fostering a milieu conducive to β-cell function[11]. Collectively, these features highlight GLP-2 as a mechanistically complementary partner to GLP-1, emphasizing β-cell preservation, insulin sensitization, and anti-inflammatory effects rather than insulinotropic action alone. Nevertheless, potential GLP-2–mediated increases in nutrient absorption, glucagon secretion, and intestinotrophic effects remain important considerations for therapeutic design.

In this study, we aim to demonstrate that simultaneous activation of GLP-1 and GLP-2 receptors can overcome the limitations of current incretin-based therapies in advanced type 2 diabetes. To this end, we use the *db/db* mouse model, which captures uncontrolled hyperglycemia, β-cell failure, and catabolic weight loss, to evaluate the metabolic and mechanistic effects of a potency-optimized GLP-1/GLP-2 bispecific agonist, PG-102. We further investigate how dual receptor engagement influences β-cell preservation, peripheral glucose handling, and receptor trafficking compared with clinically available incretin agonists and monospecific Fc fusions. Finally, we translate these findings into a phase 1 multiple ascending dose study to characterize the pharmacokinetics, pharmacodynamics, and tolerability of PG-102 in humans. Collectively, this integrated preclinical-to-clinical approach establishes the rationale for PG-102 as a differentiated therapeutic candidate that decouples glycemic control from weight loss in advanced type 2 diabetes.

## Results

### Structural design and bioactivity of PG-102

We previously developed PG-105, a prototype GLP-1/GLP-2 Fc fusion protein based on the Neo Tri-ImmunoGlobulin (NTIG) platform, and demonstrated in a metabolic dysfunction-associated steatohepatitis (MASH) model that dual receptor activation improved glucose homeostasis, strengthened intestinal barrier integrity, and reduced hepatic steatosis and fibrosis[12]. While these findings established the therapeutic potential of GLP-1/GLP-2 co-agonism, PG-105 exhibited excessive GLP-2R activity, leading to marked increases in intestinal mass and gallbladder volume (Supplementary Fig. 1), raising concerns about potential trophic effects upon chronic exposure. To mitigate these liabilities, a panel of nine GLP-1/GLP-2 NTIG Fc constructs with graded GLP-2R potencies was generated, from which PG-102 was selected as the optimal candidate. PG-102 preserved the favorable gut barrier effects—reflected by reduced serum endotoxin levels—while showing attenuated increases in intestinal parameters compared with PG-105, and notably, no statistically significant gallbladder enlargement, defining a more balanced and clinically viable pharmacologic profile.

To describe the specific design, PG-102 is a "bivalent" agonist in which GLP-1 and GLP-2 analogs are fused to opposite arms of the NTIG Fc platform (Fig. 1a). The NTIG construct, based on the hyFc backbone[13], incorporates T250Q and M428L mutations in the FcRn binding region to enhance binding affinity and prolong half-life,

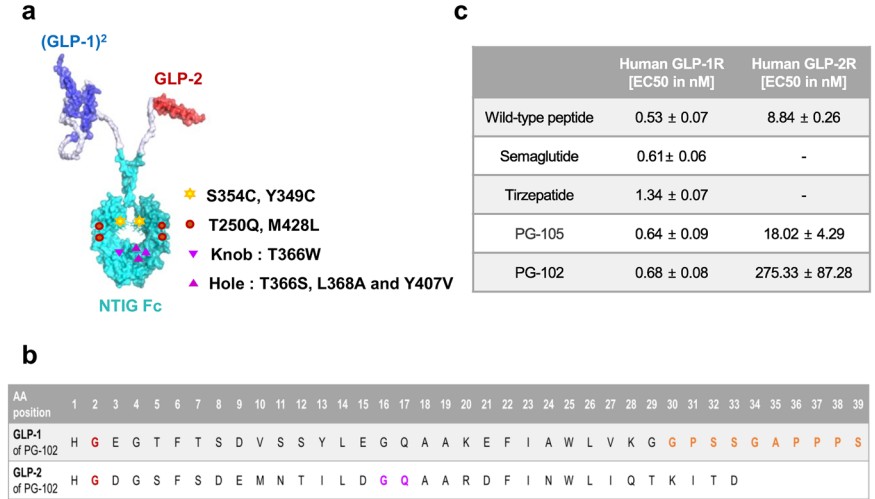

**a**

**(GLP-1)²**  **GLP-2**

★ S354C, Y349C
● T250Q, M428L
▼ Knob : T366W
▲ Hole : T366S, L368A and Y407V

**NTIG Fc**

**c**

| | Human GLP-1R [EC50 in nM] | Human GLP-2R [EC50 in nM] |
|---|---|---|
| Wild-type peptide | 0.53 ± 0.07 | 8.84 ± 0.26 |
| Semaglutide | 0.61 ± 0.06 | - |
| Tirzepatide | 1.34 ± 0.07 | - |
| PG-105 | 0.64 ± 0.09 | 18.02 ± 4.29 |
| PG-102 | 0.68 ± 0.08 | 275.33 ± 87.28 |

**b**

| AA position | 1 | 2 | 3 | 4 | 5 | 6 | 7 | 8 | 9 | 10 | 11 | 12 | 13 | 14 | 15 | 16 | 17 | 18 | 19 | 20 | 21 | 22 | 23 | 24 | 25 | 26 | 27 | 28 | 29 | 30 | 31 | 32 | 33 | 34 | 35 | 36 | 37 | 38 | 39 |
|---|---|---|---|---|---|---|---|---|---|---|---|---|---|---|---|---|---|---|---|---|---|---|---|---|---|---|---|---|---|---|---|---|---|---|---|---|---|---|---|
| GLP-1 of PG-102 | H | G | E | G | T | F | T | S | D | V | S | S | Y | L | E | G | Q | A | A | K | E | F | I | A | W | L | V | K | G | G | P | S | S | G | A | P | P | P | S |
| GLP-2 of PG-102 | H | G | D | G | S | F | S | D | E | M | N | T | I | L | D | G | Q | A | A | R | D | F | I | N | W | L | I | Q | T | K | I | T | D | | | | | | |

**Fig. 1 | Structural design and receptor pharmacology of PG-102. a** Schematic representation of PG-102, a heterodimeric (GLP-1)₂/GLP-2 NTIG Fc fusion protein. GLP-1 moieties are shown in blue, GLP-2 in red, and the NTIG Fc domain in cyan. Engineered mutations are indicated as follows: additional disulfide bond formation (yellow star-shaped markers; S34C, Y43C), enhanced FcRn binding affinity (red circular markers; T250Q, M428L), knob heterodimerization (purple downward-pointing triangular markers; T366W), and hole heterodimerization (purple upward-pointing triangular markers; T366S, L368A, Y407V). **b** Amino acid sequences of the GLP-1 and GLP-2 domains. The A2G substitution is shown in red. In GLP-1, the 30th residue was replaced with the nine C-terminal residues of exenatide (highlighted in orange). In GLP-2, L17G and A18Q substitutions are indicated in purple. **c** In vitro receptor activation assays for human GLP-1 and GLP-2 receptors. HEK-293 cells stably expressing GLP-1R or GLP-2R were treated with native peptides, semaglutide, tirzepatide, PG-105, or PG-102, and intracellular cAMP accumulation was quantified (*n* = 3). Data are presented as EC₅₀ values (mean ± SD). NTIG Neo Tri- ImmunoGlobulin, cAMP cyclic adenosine monophosphate, SD standard deviation. Source data are provided as a Source data file.

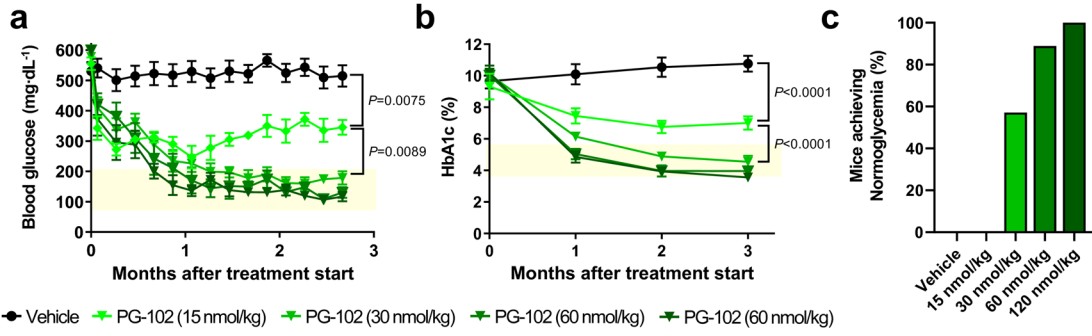

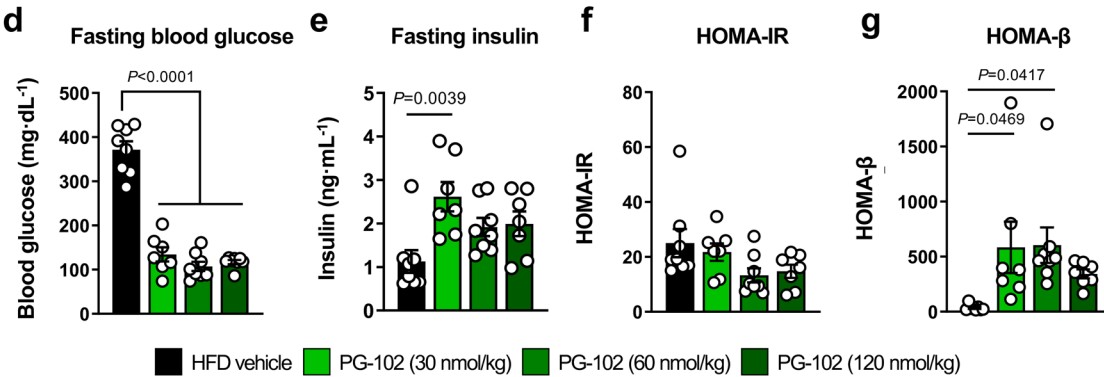

**Fig. 2 | Glycemic control and metabolic effects of PG-102 in a *db/db* mouse model of severe hyperglycemia.** Male *db/db* mice (14 weeks of age; baseline HbA1c > 10%) were randomized to receive vehicle (*n* = 14) or PG-102 administered subcutaneously every 3 days for 12 weeks at doses of 15 (*n* = 7), 30 (*n* = 15), 60 (*n* = 9), or 120 (*n* = 9) nmol/kg. **a** Longitudinal changes in non-fasting blood glucose. **b** Longitudinal changes in HbA1c. **c** Proportion of mice achieving normoglycemia at study endpoint (blood glucose <200 mg/dL). End-of-study assessments, including **d** fasting blood glucose, **e** fasting insulin, **f** homeostasis model assessment of insulin resistance (HOMA-IR), and (**g**) homeostasis model assessment of β-cell function (HOMA-β). Data are shown as mean ± SEM. Statistical significance was evaluated using two-way ANOVA with Tukey's post hoc test for longitudinal analyses (**a**–**c**) and one-way ANOVA with Tukey's post hoc test for endpoint comparisons (**d**–**g**). n.d. not detected, SEM standard error of the mean. Source data are provided as a Source data file.

ensuring sustained in vivo activity. Additionally, targeted mutations—S354C and T366W in one segment and Y349C, T366S, L368A, and Y407V in another—facilitate heterodimeric assembly through a knob-and-hole mechanism[14]. The GLP-1 analog was engineered with an A2G mutation to prevent DPP-4 cleavage and further modified by replacing its last two amino acids with ten C-terminal residues from exenatide at position 30 to enhance receptor binding affinity[15,16]. A tandem GLP-1 construct linked by a GS linker was employed to facilitate post-purification identification of the heterodimeric GLP-1/GLP-2-NTIG Fc structure and to enhance GLP-1 activity[17]. The GLP-2 analog carried an A2G mutation to prevent DPP-4 cleavage, together with N17G and L18Q substitutions to improve structural stability and reduce aggregation (Fig. 1b)[18].

Receptor potency of PG-102 was evaluated using cAMP production assays in CHO-K1 cells overexpressing GLP-1R and HEK293 cells overexpressing GLP-2R. PG-102 exhibited an EC$_{50}$ of 0.68 nM for GLP-1R activation, representing 78% of native GLP-1 potency (0.68 nM vs. 0.53 nM), ~90% of semaglutide (0.68 nM vs. 0.61 nM), and nearly twice that of tirzepatide (0.68 nM vs. 1.34 nM). For GLP-2R activation, PG-102 retained an EC$_{50}$ of 275.33 nM, corresponding to only 3.4% of native GLP-2 potency (275.33 nM vs. 8.84 nM), reflecting deliberate attenuation of GLP-2R activity (Fig. 1c). Given the structural homology across class-B GPCRs, we next profiled potential off-target activation at human GIPR and GCGR using matched cAMP assays. PG-102 showed no detectable agonist activity at either receptor, whereas native ligands (GIP, glucagon) and benchmark multi-agonists (tirzepatide, retatrutide) produced robust responses, confirming assay sensitivity.

Together, these data indicate that PG-102 is selective for GLP-1R and GLP-2R (Supplementary Fig. 2).

Importantly, a 26-week repeat-dose toxicity study of PG-102 demonstrated the absence of hyperplasia not only in thyroid C cells but also throughout the gastrointestinal tract. In contrast, rodent repeat-dose studies of GLP-1–based drugs such as semaglutide and tirzepatide have reported thyroid C cell hyperplasia, while monkey studies of the GLP-2 analog teduglutide have shown gastrointestinal hyperplasia (Supplementary Table 1). These findings establish PG-102 as a bispecific GLP-1/GLP-2 receptor agonist engineered to balance efficacy with improved safety margins.

## Dose-dependent glycemic effects of PG-102 in *db/db* mice

We conducted a dose-dependency study to determine the optimal therapeutic dose of PG-102 using a 14-week *db/db* mouse model, which recapitulates features of advanced T2D, including uncontrolled hyperglycemia (baseline HbA1c > 10%), catabolic weight loss, and progressive β-cell dysfunction[19]. PG-102 demonstrated a clear dose-dependent reduction in both non-fasting blood glucose and HbA1c levels over a 3-month treatment period (Fig. 2a, b). The 15 nmol/kg dose modestly reduced glucose and HbA1c but failed to induce normoglycemia (defined as blood glucose <200 mg/dL)[20], with 0% of mice achieving this benchmark (Fig. 2c). In contrast, the 30 nmol/kg dose enabled over half of treated mice (57.1%) to achieve normoglycemia, whereas 60 and 120 nmol/kg induced normoglycemia in nearly all (88.9%) and all (100%) treated mice, respectively, demonstrating substantial improvement in glycemic control relative to the

subtherapeutic 15 nmol/kg dose. Consequently, the 15 nmol/kg dose was excluded from further analyses due to its suboptimal performance.

In subsequent analyses, PG-102 significantly reduced fasting blood glucose levels across all effective doses (Fig. 2d) and increased fasting insulin levels at 30 nmol/kg (Fig. 2e), consistent with improved glucose metabolism driven by enhanced β-cell function. HOMA-IR values demonstrated a downward trend, suggesting improved insulin sensitivity, though statistical significance was not observed (Fig. 2f). HOMA-β scores were increased across PG-102 treatment groups, reaching statistical significance at 30 and 60 nmol/kg doses, suggesting enhanced β-cell preservation and function (Fig. 2g). The plateau in HOMA-β across 30–120 nmol/kg suggests that β-cell preservation reached a functional ceiling once normoglycemia was established, consistent with saturating efficacy at higher doses. Together, these data establish 30 nmol/kg as the minimal efficacious dose that achieves durable normoglycemia while preserving β-cell function, and it was therefore selected for subsequent head-to-head comparative studies.

### Comparative efficacy of PG-102, semaglutide, and tirzepatide

We next evaluated the efficacy of PG-102 against semaglutide and tirzepatide. A matching dose of 30 nmol/kg was selected for semaglutide to align with PG-102, whereas 15 nmol/kg was chosen for tirzepatide, reflecting prior reports that this lower dose yields metabolic efficacy comparable to 30 nmol/kg semaglutide in rodents[21,22]. Vehicle-treated *db/db* mice exhibited persistent severe hyperglycemia (non-fasting blood glucose ~600 mg/dL; HbA1c > 10%). Semaglutide and tirzepatide reduced blood glucose by ~200 mg/dL and HbA1c by ~2% at 2 months, but their effects waned by the third month. In contrast, PG-102 achieved a ~450 mg/dL glucose reduction and ~5% HbA1c decrease by 2-months, restoring normoglycemia and maintaining efficacy throughout the 3-month study (Fig. 3a, b).

Body weight analysis highlighted an important distinction. Vehicle-treated mice showed rapid weight loss, reflecting the catabolic state of uncontrolled diabetes. Semaglutide and tirzepatide partially mitigated this decline, whereas PG-102 more effectively preserved body weight, countering the metabolic deterioration (Fig. 3c).

To further characterize these effects, we examined pancreatic islets histologically. Vehicle-treated mice displayed severe islet disorganization and loss of insulin-positive β-cells. Semaglutide partially improved islet morphology, whereas tirzepatide offered little benefit. By contrast, PG-102 markedly preserved β-cell area and islet architecture, consistent with its robust glycemic effects (Fig. 3d, g, h). PG-102 also reduced α-cell area compared with vehicle and showed a trend toward lower levels than semaglutide or tirzepatide, suggesting decreased glucagon production (Fig. 3e, i). Moreover, Ki67-insulin double staining revealed that PG-102 significantly maintained the proportion of proliferating β-cells, supporting preservation of regenerative potential (Fig. 3f, j). We additionally assessed inflammatory involvement in peri-pancreatic adipose tissue, a visceral fat depot enriched in leukocytes and implicated as a source of mediators affecting β-cell dysfunction[23,24]. Although CD45+ staining was also performed on pancreatic sections, intact islets were largely absent in vehicle-treated mice, precluding meaningful quantification. Accordingly, peri-pancreatic fat served as the most reliable site to evaluate inflammation, where vehicle-treated mice displayed abundant CD45+ cells that were substantially reduced by all incretin-based treatments. Notably, PG-102 showed the lowest CD45+ area among groups, suggesting a possible trend toward reduced local inflammation compared to semaglutide or tirzepatide (Supplementary Fig. 3).

Finally, because tirzepatide's GIPR agonism is attenuated in rodents[25], we performed an additional study in *db/db* mice with moderate hyperglycemia (non-fasting blood glucose 500–600 mg/dL;

9% < HbA1c < 10%), using equal doses of PG-102 and tirzepatide (30 nmol/kg). In this setting, PG-102 consistently achieved superior glucose lowering, HbA1c reduction, body weight preservation, and β-cell area maintenance compared with tirzepatide (Supplementary Fig. 4), reinforcing its advantage across different stages of disease severity. Given the species-dependent pharmacology of tirzepatide, these findings should be interpreted as comparative references rather than direct predictors of clinical efficacy, yet they highlight the differentiated profile of PG-102.

### In vitro mechanisms of PG-102 versus incretin comparators

We next investigated the β-cell–centric mechanisms underlying PG-102–mediated glycemic control. In perifusion assays using isolated islets from normoglycemic mice, PG-102 augmented glucose-stimulated insulin secretion (GSIS; 3 mM → 11 mM), compared with vehicle, and trended higher than semaglutide and tirzepatide, although without statistical significance (Fig. 4a). While semaglutide and tirzepatide induced comparable GSIS despite differing GLP-1R potency, this likely reflects the limited contribution of GIPR agonism in rodents[25]. Importantly, PG-102 achieved numerically greater insulin secretion than semaglutide despite equivalent GLP-1R potency, suggesting possible additive contributions from GLP-2R co-activation.

We next assessed β-cell protection under diabetogenic stress using the streptozotocin (STZ)-induced cytotoxicity in INS-1 cells, which co-express GLP-1R and GLP-2R (Supplementary Fig. 5a). PG-102 significantly preserved β-cell viability relative to vehicle, semaglutide, and tirzepatide (Fig. 4b). Under high-glucose challenge (16.8 mM), PG-102 also promoted robust insulin release, surpassing semaglutide and far exceeding vehicle control (Fig. 4c). At the molecular level, PG-102 significantly upregulated β-cell transcription factors *Pdx1* and *Neurod1* ($p < 0.05$), whereas *MafA* showed an upward, non-significant trend; all three factors are essential for insulin gene expression, β-cell identity, and survival (Supplementary Fig. 6). These findings indicate that PG-102 protects β-cells from cytotoxic injury while enhancing their insulinotropic function by reinforcing key transcriptional programs.

Finally, to explore systemic metabolic implications, we examined the effect of PG-102 on peripheral glucose disposal. Both 3T3-L1 adipocytes and L6-GLUT4myc myotubes expressed GLP-1R and GLP-2R, as confirmed by flow cytometry (Supplementary Fig. 5b, c). In glucose uptake assays, PG-102 markedly enhanced glucose uptake compared with semaglutide or tirzepatide (Fig. 4d, e), highlighting its superior potential to improve peripheral insulin sensitivity.

### Bivalent receptor engagement underlying PG-102 efficacy

To directly evaluate the contribution of dual receptor engagement, we compared monospecific GLP-1-NTIG Fc, GLP-2-NTIG Fc, their co-administration, and the bispecific PG-102 in an intraperitoneal glucose tolerance test (ipGTT) in normoglycemic mice. At GLP-1R–matched dosing, PG-102 enhanced glucose clearance more effectively than GLP-1-NTIG Fc or the co-administered combination, whereas GLP-2-NTIG Fc alone exerted minimal effect (Fig. 5a). Given the ceiling effect of ipGTT in healthy mice, these differences likely underestimate the functional advantage of PG-102, but nonetheless support in vivo synergy arising from simultaneous GLP-1R and GLP-2R activation.

We next assessed β-cell protection under diabetogenic stress. In STZ-challenged INS-1 cells, PG-102 consistently outperformed monospecific controls and equaled or surpassed the co-treatment of GLP-1-NTIG Fc and GLP-2-NTIG Fc across glucose-stimulated insulin secretion and preservation of key β-cell transcription factors, including *MafA*, *Pdx1*, and *Neurod1* (Fig. 5b and Supplementary Fig. 7a–c). PG-102 also significantly attenuated STZ-induced *Tnf-α* expression, exceeding the effect of GLP-1-NTIG Fc and GLP-2-NTIG Fc co-treatment, consistent with a protective and anti-inflammatory role in β-cell stress (Supplementary Fig. 7d).

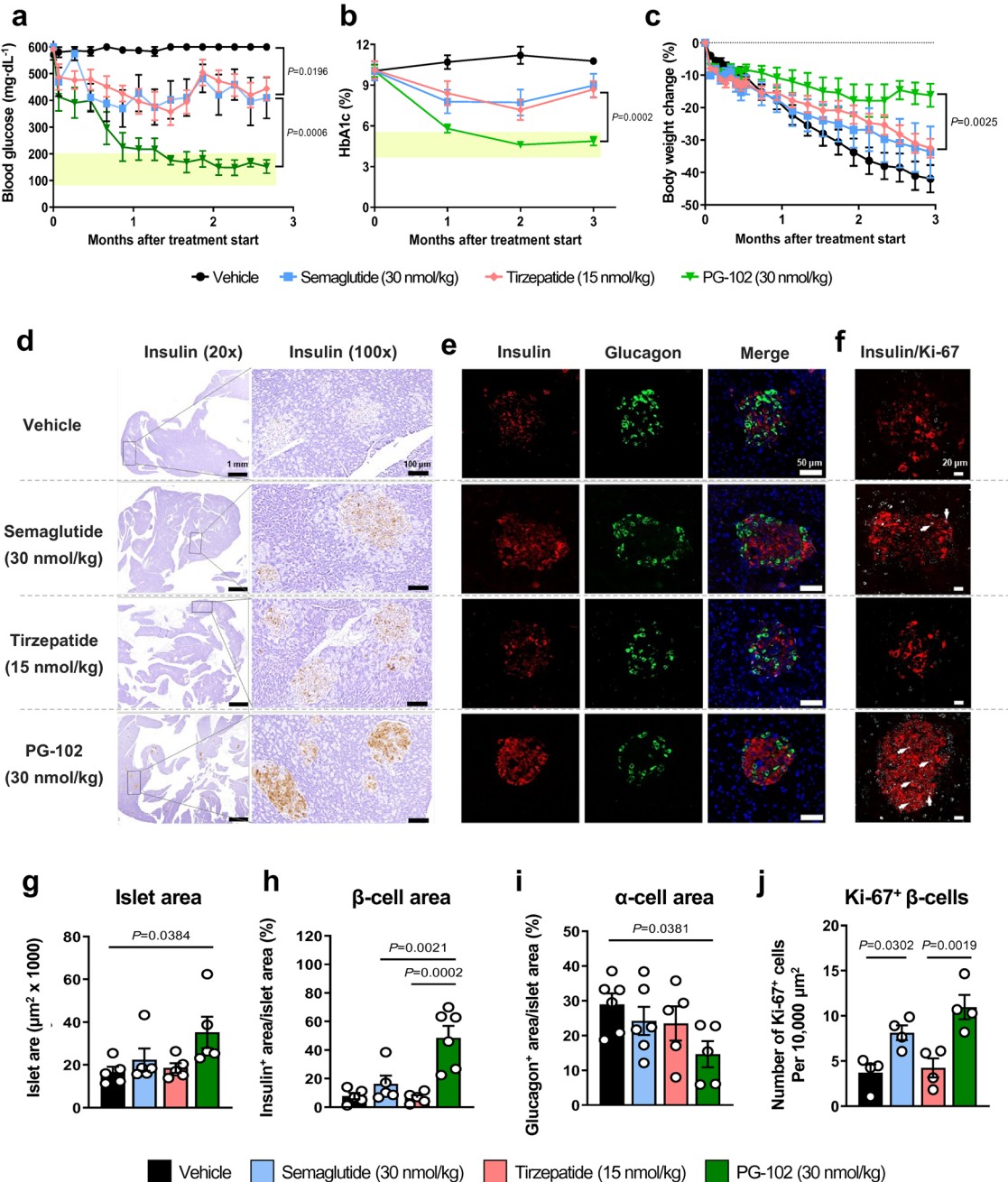

**Fig. 3 | Comparative efficacy of PG-102, semaglutide, and tirzepatide on glycemic control and pancreatic islet preservation in *db/db* mice.** Male *db/db* mice (14 weeks of age) were randomized to receive PG-102 (30 nmol/kg), semaglutide (30 nmol/kg), tirzepatide (15 nmol/kg), or vehicle, administered subcutaneously every 3 days for 12 weeks. Longitudinal changes in (**a**) non-fasting blood glucose, (**b**) HbA1c, and (**c**) body weight are shown as mean ± SEM (*n* = 9). Missing values at certain time points reflect unsuccessful or insufficient blood collection. Representative pancreatic histology from mice in each group: (**d**) insulin immunostaining (*n* = 5 mice), (**e**) dual immunofluorescence for insulin and glucagon (*n* = 5 mice), and

(**f**) dual immunofluorescence for insulin and Ki-67 (*n* = 4 mice). Quantitative analyses of pancreatic islet composition and proliferation: (**g**) islet area, (**h**) β-cell area, (**i**) α-cell area (*n* = 5 mice each), and (**j**) Ki-67+ cells per 10,000 μm² (*n* = 4 mice). Each data point represents one mouse (biological replicate), with quantification performed using one representative section per mouse. Statistical significance was evaluated using two-way ANOVA with Tukey's post hoc test for longitudinal analyses (**a**–**c**) and one-way ANOVA with Tukey's post hoc test for endpoint comparisons (**g**–**j**). SEM, standard error of the mean. Source data are provided as a Source data file.

Finally, to explore effects beyond islets, we investigated peripheral glucose metabolism using differentiated 3T3-L1 adipocytes and L6-GLUT4myc myotubes. PG-102 elicited the greatest increase in glucose uptake relative to monospecific Fc fusions or their co-treatment (Fig. 5c, d), indicating superior enhancement of peripheral glucose disposal. Mechanistic dissection with receptor-selective antagonists revealed that this effect was strongly GLP-

2R-dependent, as it was abrogated by GLP-2 (3–33), whereas exendin (9–39) produced only modest inhibition (Supplementary Fig. 8). Although PG-102 displayed only ~3% of the native GLP-2R potency in cAMP assays (Fig. 1c), these blockade studies demonstrated that its glucose uptake effects were predominantly GLP-2R-driven, underscoring a functional contribution not apparent from potency data alone.

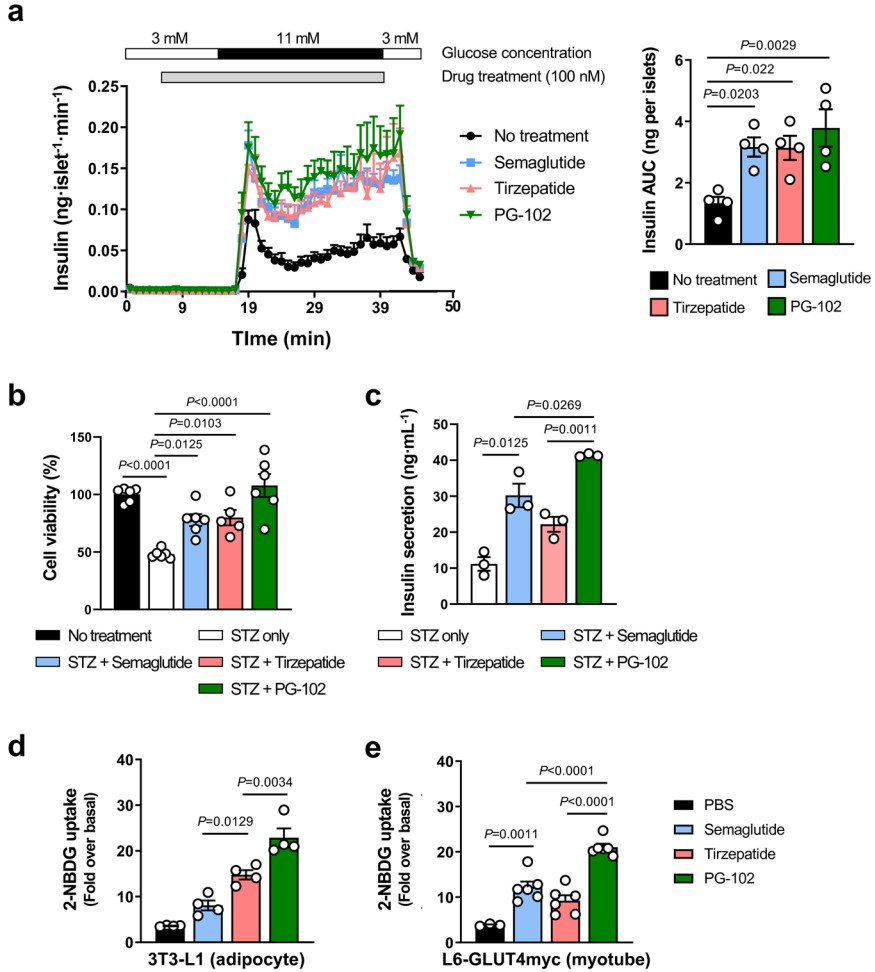

**Fig. 4 | Comparative actions of PG-102, semaglutide, and tirzepatide on β-cell function, cytoprotection, and peripheral glucose uptake. a** Glucose-stimulated insulin secretion from isolated mouse islets assessed using a perifusion system. Insulin secretion was monitored following a glucose shift (3–11 mM) in the presence of semaglutide, tirzepatide, or PG-102 (100 nM, left panel). Quantification of insulin secretion was performed by area under the curve (AUC; right panel). $n = 4$ independent islet preparations derived from different mice; each preparation was considered a biological replicate. **b, c** Cytoprotective and functional effects in INS-1 pancreatic β-cells exposed to streptozotocin (STZ). Cells were pretreated with the indicated agents (300 nM, 24 h) followed by exposure to STZ (10 nM, 3 h). **b** Cell viability assessed by MTS assay ($n = 6$ independent culture wells per group). (**c**) After 2-h glucose starvation, insulin secretion was measured following 30 min stimulation with 16.8 mM glucose using an ultrasensitive ELISA ($n = 3$ independent culture wells per group). Glucose uptake assessed by 2-NBDG uptake assay in differentiated 3T3-L1 adipocytes ($n = 4$ independent experiments) (**d**) and L6-GLUT4myc myotubes ($n = 6$ independent experiments) (**e**) following treatment with the indicated agents (300 nM). Data are presented as mean ± SEM. Statistical significance was evaluated using one-way ANOVA with Tukey's post hoc test. AUC, area under the curve; STZ, streptozotocin; MTS, [3-(4,5-dimethylthiazol-2-yl)-5-(3-carboxymethoxyphenyl)-2-(4-sulfophenyl)-2H-tetrazolium]; ELISA, enzyme-linked immunosorbent assay; 2-NBDG, 2-(N-(7-nitrobenz-2-oxa-1,3-diazol-4-yl)amino)-2-deoxyglucoseSEM, standard error of the mean. Source data are provided as a Source data file.

Although these functional studies highlighted a predominant role of GLP-2R in mediating PG-102's metabolic effects, the underlying receptor-level mechanism remained to be clarified. To further investigate receptor dynamics, we performed confocal imaging of GLP-1R and GLP-2R trafficking in HEK293 cells co-expressing both receptors. GLP-1–NTIG Fc and the GLP-1/GLP-2 co-treatment triggered rapid GLP-1R internalization (peak ~30–60 min), whereas GLP-2R signals remained diffusely distributed across the membrane and cytoplasm (Supplementary Fig. 9a, b). In contrast, PG-102 did not induce rapid receptor loss from the surface but maintained clustered receptor patterns with gradual redistribution over 120 min, suggesting *cis*-co-engagement of GLP-1R and GLP-2R that supports coordinated trafficking and sustained signaling.

## Phase 1 MAD trial of PG-102: safety, PK and PD evaluation
A phase 1 multiple ascending dose (MAD) study (NCT06309667) was conducted to evaluate the safety, tolerability, pharmacokinetic (PK), and pharmacodynamic (PD) profiles of PG-102 in healthy adults with a body mass index (BMI) of 25–30 kg/m². Safety and tolerability were predefined as the primary endpoint, assessed by treatment-emergent adverse events (TEAEs), while secondary endpoints included PK and PD assessments. Exploratory endpoints comprised inflammation-related biomarkers such as plasma high-sensitivity C-reactive protein (hsCRP).

This randomized, double-blind, placebo-controlled trial included 24 participants aged 18–65 years, who were randomized in a 6:2 ratio to receive PG-102 or placebo within three dose cohorts: Cohort 1 (15 mg), Cohort 2 (30 mg), and Cohort 3 (30/60 mg, with an escalating regimen of 30/30/60/60/60 mg), administered weekly. A schematic overview of the trial design and allocation is provided in Fig. 6 and Supplementary Table 2. Baseline characteristics were well-balanced (mean age $30.9 \pm 6.7$ years, BMI $27.0 \pm 1.2$ kg/m², body weight of $80.6 \pm 7.0$ kg), with males comprising ~92% of the study population (Table 1). The 15 mg starting dose was informed by the preceding

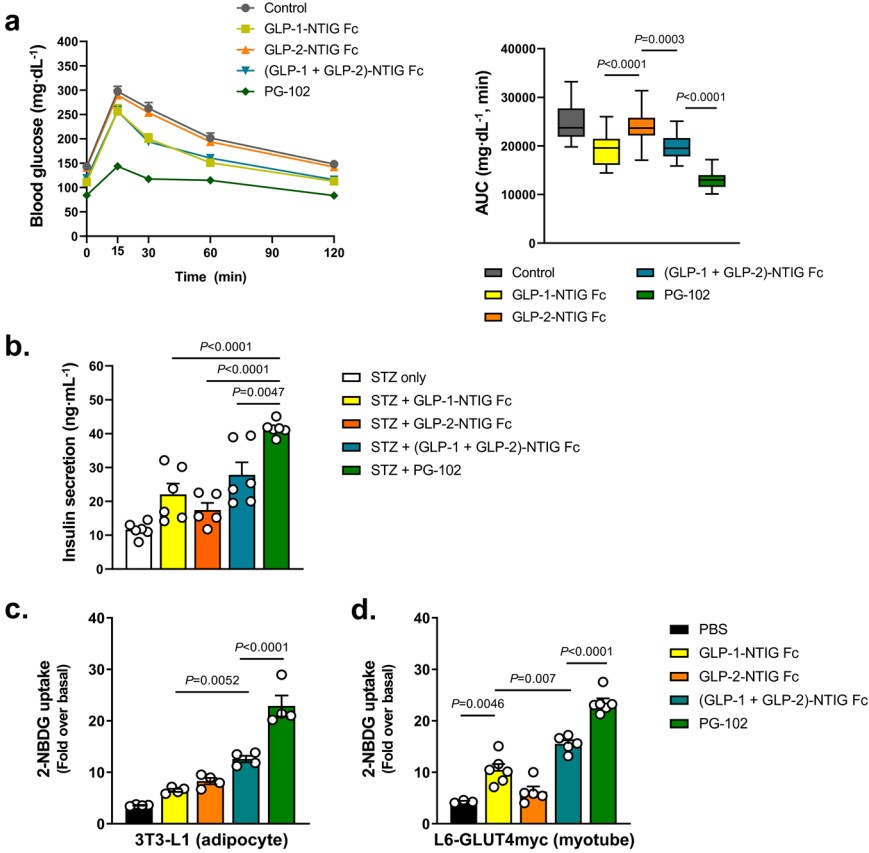

**Fig. 5 | Comparative efficacy of PG-102 versus GLP-1-NTIG Fc, GLP-2-NTIG Fc, and their combination. a** Intraperitoneal glucose tolerance test (ipGTT) in 10-week-old C57BL/6 J mice. Mice received a single subcutaneous dose of the indicated agents (15 nmol/kg). After 24 h, including a 6-h fast, an ipGTT was performed (2 g glucose/kg body weight), and blood glucose monitored for 120 min ($n = 19$ per group, left). Glucose clearance was quantified as area under the curve (AUC; right). Box plots indicate the median (center line), interquartile range (box; 25th–75th percentiles), and minimum to maximum values (whiskers). **b** Glucose-stimulated insulin secretion (GSIS) in INS-1 pancreatic β-cells. Cells were pretreated for 24 h with GLP-1-NTIG Fc, GLP-2-NTIG Fc, their combination, or PG-102 (300 nM), followed by exposure to STZ (5 mM, 3 h). GSIS was assessed after equilibration at 2.4 mM glucose and stimulation with 16.8 mM for 1 h ($n = 6$ independent culture wells per group). Insulin-stimulated glucose uptake assessed by 2-NBDG uptake assay in differentiated 3T3-L1 adipocytes ($n = 4$ independent experiments) (**c**) and L6-GLUT4myc myotubes ($n = 6$ independent experiments) (**d**) following treatment with the indicated agents (300 nM). Data in (**b**–**d**) are presented as mean ± SEM. Statistical significance was evaluated using one-way ANOVA with Tukey's post hoc test. ipGTT intraperitoneal glucose tolerance test, AUC area under the curve, GSIS glucose-stimulated insulin secretion, STZ streptozotocin, 2-NBDG 2-(N-(7-nitro-benz-2-oxa-1,3-diazol-4-yl)amino)-2-deoxyglucose, SEM standard error of the mean. Source data are provided as a Source data file.

single ascending dose (SAD) study and a 4-week toxicology study in cynomolgus monkeys, which established a NOAEL of 6 mg/kg, providing a wide safety margin (Supplementary Table 3). Participant recruitment was conducted between March 2024 and July 2024, and follow-up was completed by September 2024.

Safety assessments across all dose cohorts demonstrated that PG-102 was well-tolerated. TEAEs occurred in 19 participants (79.2%), with treatment-related AEs reported in 18 participants (75.0%; Table 2). No serious adverse events (SAEs) or discontinuations due to treatment-related AEs occurred. GI events, typical of GLP-1RAs[26], represented the majority of TEAEs: dyspepsia (41.7%), diarrhea (20.8%), nausea (25.0%), and vomiting (12.5%). Dyspepsia showed a dose-related increase (33.3–83.3% across cohorts). Vomiting was infrequent (0–33.3%) and absent in the placebo and 15 mg groups. Nausea incidence was consistent at 33.3% across all PG-102 doses, while diarrhea showed no clear dose relationship and was comparable to placebo (16.7%). All GI-related events were mild to moderate, with no evidence of worsening over time.

PK analyses following the final dose demonstrated dose-proportional exposure and consistent elimination kinetics (Fig. 7a and Supplementary Table 4). $T_{max}$ ranged from 48 h (30 mg) to 72 h (15 mg), with a median of 60 h in the 30/60 mg cohort, reflecting delayed absorption. $AUC_{0-inf}$ increased proportionally with dose (259 μg h/mL at 15 mg; 660 μg h/mL at 30/60 mg). The terminal half-life ($t_{1/2}$) was 107–118 h across cohorts, and clearance ($CL_{ss}/F$) remained low (0.06–0.11 L/h), confirming slow systemic elimination.

PD effects were evaluated by oral glucose tolerance test (OGTT) on day 30, 1 day post-final dose. In the placebo, glucose peaked at 60 min, whereas PG-102 shifted the peak earlier (30 min) and accelerated the return to baseline (Fig. 7b, left panel). All PG-102 cohorts showed numerical reductions in glucose $AUC_{0-2h}$ compared with placebo, with decreases of 26.5%, 24.9%, and 19.8% in the 30 mg, 30/60 mg, and 15 mg groups, respectively (Fig. 6b, right panel). Exploratory exposure–response analysis suggested a trend toward greater glucose tolerance improvements with higher systemic exposure (Supplementary Fig. 10). Exploratory biomarkers, including body weight, composition, and inflammatory indices, showed no changes in this short healthy-volunteer study (Supplementary Tables 5 and 6).

## Discussion
Metformin is widely recommended as the first-line treatment for type 2 diabetes; however, in cases of severe hyperglycemia (HbA1c level >10%), insulin remains the standard therapy due to its rapid and potent glucose-lowering effects[27]. Yet insulin use is often limited by a high risk

of hypoglycemia and pronounced glucose variability. In contrast, GLP-1RAs provide an effective alternative, achieving robust glycemic control with a lower risk of hypoglycemia and reduced variability[28]. Nevertheless, even the most effective marketed incretin, the GLP-1/GIP dual agonist tirzepatide, produces a maximum HbA1c reduction of ~2.6%, with fewer than half of participants in the SURPASS trials reaching normoglycemia (HbA1c < 5.7%)[29]. This highlights the unmet need for therapies capable of inducing and sustaining optimal glycemic control in patients with severe hyperglycemia.

In this study, PG-102, a bispecific GLP-1/GLP-2 receptor agonist, demonstrated marked glycemic efficacy, with an ~5% HbA1c reduction in severely hyperglycemic *db/db* mice. The reductions observed with semaglutide (~1.7%) and tirzepatide (~2.5%) in the same model were concordant with their clinical trial outcomes, supporting the translational validity of this model. Accordingly, the robust HbA1c reduction achieved by PG-102, together with restoration of normoglycemia without hypoglycemia, suggests the potential for this superior efficacy to extend into clinical practice.

PG-102's robust anti-hyperglycemic efficacy appears to arise from multiple, synergistic mechanisms acting across β-cells, peripheral tissues, and receptor pharmacology. First, PG-102 sustains β-cell health by integrating the complementary biology of GLP-1 and GLP-2. While GLP-1 supports proliferation and limits apoptosis, GLP-2 enhances survival under metabolic stress and protects against cellular damage[1,8]. Consistent with this, PG-102 preserved β-cell transcriptional programs (*MafA*, *Pdx1*, *Neurod1*) and mitigated diabetogenic stress, including suppression of *Tnf-α* induction. These findings highlight a dual benefit —direct protection of β-cell mass and an anti-inflammatory signature— that is central to long-term glycemic stability. Second, PG-102 augments systemic glucose disposal through enhanced peripheral glucose uptake. In murine adipocytes and rat myotubes, PG-102 promoted markedly greater glucose uptake than monospecific or co-treatment controls, with receptor blockade studies revealing a predominant contribution from GLP-2R. This indicates that PG-102 improves glucose regulation not only by preserving insulin secretion but also by enhancing insulin sensitivity in peripheral tissues. Consistent with this, recent studies have demonstrated that GLP-2 can directly stimulate glucose uptake in human subcutaneous adipocytes, independent of insulin signaling[10]. The co-expression of GLP-1R and GLP-2R in adipose tissue and other metabolic compartments such as the gut and brain underscores the translational relevance of this dual axis[10,30–33]. Third, PG-102 exerts anti-inflammatory effects that may further reinforce metabolic stability. In *db/db* mice, peri-pancreatic adipose tissue—an immune-rich visceral depot closely linked to β-cell stress[24]—showed marked immune infiltration under vehicle treatment, which was mitigated by all incretin-based therapies. Among these, PG-102 tended to show the greatest reduction in CD45+ cell infiltration, suggesting a potential anti-inflammatory contribution from GLP-2R engagement. This effect, together with the observed suppression of TNFα induction under β-cell stress, suggests that PG-102 may extend the metabolic benefits of incretin therapy by attenuating low-grade inflammation at both endocrine and peripheral levels. Consistent with a previous report that GLP-2 modulates macrophage polarization and inflammatory tone within human islets[11], these findings support a broader role for GLP-2 biology in maintaining metabolic tissue homeostasis. Finally, at the level of receptor pharmacology, PG-102 capitalizes on bivalent *cis* co-engagement of GLP-1R and GLP-2R to achieve biological efficacy that is not readily predicted by isolated receptor potency measurements. Although PG-102 exhibits attenuated intrinsic potency at GLP-2R in conventional in vitro assays, growing evidence indicates that spatially constrained, *cis*-acting engagement of two receptors on the same cell surface can functionally amplify downstream signaling through enhanced receptor proximity, coordinated trafficking, and prolonged receptor residence time. Recent studies of *cis*-targeting bispecific antibodies have shown that bivalent receptor engagement

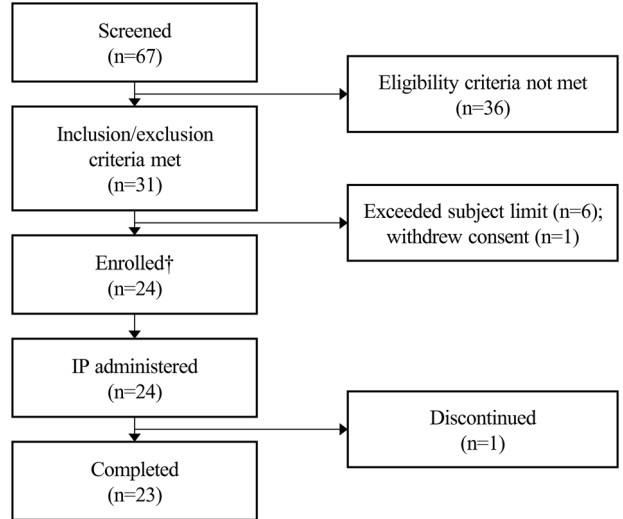

**Fig. 6 | Subject disposition flow chart in the phase 1 multiple ascending dose (MAD) study.** A total of 67 individuals were screened, with 31 meeting the inclusion and exclusion criteria. Of these, 24 participants were enrolled (eight per cohort; randomized 6:2 to PG-102 or placebo) and received the investigational product (IP; PG-102 or placebo). Reasons for exclusion included ineligibility ($n = 36$), exceeding the subject limit ($n = 6$), and consent withdrawal prior to enrollment ($n = 1$). One participant in the 15 mg cohort discontinued after the fourth dose due to an adverse event assessed as unrelated to the IP. The participant completed follow-up safety evaluations and was included in the safety analysis set but excluded from efficacy and pharmacokinetic analyses. †Enrolled = randomized. Source data are provided as a Source data file.

can stabilize receptor complexes and modulate endocytic behavior, resulting in in vivo efficacy that is disproportionate to in vitro affinity or potency metrics[34]. Similar principles were established in earlier bivalent cytokine agonists, such as leridistim, where dual engagement of IL-3R and G-CSFR yielded enhanced biological activity despite modest intrinsic activation of either receptor alone[35]. Consistent with these precedents, confocal trafficking analyses revealed that *cis* co-engagement by PG-102 delays GLP-1R internalization and promotes coordinated receptor trafficking, in contrast to the rapid and poorly synchronized receptor internalization observed with co-treatment of monospecific agonists. These findings provide a plausible mechanistic explanation for how partial GLP-2R engagement, when integrated with GLP-1R signaling in a bispecific format, can contribute disproportionately to therapeutic efficacy. Given the documented co-expression of GLP-1R and GLP-2R across multiple metabolically relevant cell types—including pancreatic islets, adipocytes, intestinal epithelial subsets, enteric neurons, and vagal afferent neurons[10,30–33],—this mechanism is likely to extend beyond the experimental systems examined here. While direct demonstration of GLP-2R-specific downstream signaling in human tissues remains an important area for future investigation, these mechanistic precedents support the concept that bivalent incretin agonism can achieve sustained metabolic benefit through receptor co-engagement rather than high intrinsic receptor potency alone. In summary, PG-102 unites β-cell preservation, enhanced glucose disposal, anti-inflammatory activity, and sustained receptor signaling—providing a mechanistic rationale for why GLP-2 matters in advanced T2D and highlighting its translational potential as a next-generation incretin therapy.

Unintentional weight loss is a hallmark of advanced T2D patients, typically arising from severe hyperglycemia and catabolic metabolism that shift energy utilization away from carbohydrates toward protein and fat[36]. This catabolic state accelerates muscle wasting and undermines clinical outcomes, representing a major therapeutic challenge. The *db/db* mouse model recapitulates this phenomenon: in our

**Table 1 | Baseline demographics and clinical characteristics of study participants**

| Category | PG-102 | | | Placebo (n = 6) | Total (n = 24) |
|---|---|---|---|---|---|
| | 15 mg (n = 6) | 30 mg (n = 6) | 30/60 mg (n = 6) | | |
| Age (years) | 30.3 ± 6.1 | 31.3 ± 7.8 | 28.5 ± 5.7 | 33.5 ± 7.8 | 30.9 (± 6.7) |
| Female, n (%) | 1 (16.7%) | 1 (16.7%) | 0 (0%) | 0 (0%) | 2 (8.3%) |
| Male n (%) | 5 (83.3%) | 5 (83.3%) | 6 (100%) | 6 (100%) | 22 (91.7%) |
| Body weight (kg) | 77.6 ± 3.5 | 80.0 ± 9.7 | 81.9 ± 5.2 | 82.8 ± 8.6 | 80.6 (± 7.0) |
| BMI (kg per m²) | 26.7 ± 1.2 | 27.0 ± 1.2 | 26.5 ± 0.3 | 27.7 ± 1.5 | 27.0 (± 1.2) |

Summary of demographic and baseline characteristics of participants (n = 24) across treatment groups. Data are presented as mean ± SD for continuous variables and n (%) for categorical variables. Sex was determined based on self-report at screening. Given to the very small number of female participants (n = 2), no sex-based subgroup analysis was performed.
n number of subjects, BMI body mass index.

**Table 2 | Treatment-emergent adverse events in a phase 1 multiple ascending dose study of PG-102**

| | PG-102 | | | Placebo (n = 6) | Total (n = 24) |
|---|---|---|---|---|---|
| | 15 mg (n = 6) | 30 mg (n = 6) | 30/60 mg (n = 6) | | |
| Any TEAEs | 5 (83.3) | 5 (83.3) | 5 (83.3) | 4 (66.7) | 19 (79.2) |
| Treatment-related AEs | 5 (83.3) | 5 (83.3) | 5 (83.3) | 3 (50.0) | 18 (75.0) |
| Discontinuation due to treatment-related AEs | 0 (0) | 0 (0) | 0 (0) | 0 (0) | 0 (0) |
| Gastrointestinal AEs | | | | | |
| Abdominal distension | 0 (0) | 0 (0) | 0 (0) | 1 (16.7) | 1 (4.2) |
| Constipation | 0 (0) | 3 (50.0) | 1 (16.7) | 0 (0) | 4 (16.7) |
| Diarrhea | 1 (16.7) | 2 (33.3) | 1 (16.7) | 1 (16.7) | 5 (20.8) |
| Dyspepsia | 2 (33.3) | 3 (50.0) | 5 (83.3) | 0 (0) | 10 (41.7) |
| Gastrointestinal reflux disease | 0 (0) | 1 (16.7) | 0 (0) | 0 (0) | 1 (4.2) |
| Nausea | 2 (33.3) | 2 (33.3) | 2 (33.3) | 0 (0) | 6 (25.0) |
| Vomiting | 0 (0) | 1 (16.7) | 2 (33.3) | 0 (0) | 3 (12.5) |

Data presented as n (%).
Summary of treatment-emergent adverse events (TEAEs) in participants receiving PG-102 (15 mg, 30 mg, 30/60 mg weekly; n = 6 per cohort) or placebo (n = 6). TEAEs are classified as overall events, treatment-related events, or gastrointestinal-related events. Data are presented as the number of participants experiencing at least one AE, with percentages calculated relative to the total number of participants in each treatment group. Source data are provided as a Source data file.
TEAE treatment-emergent adverse event, AE serious adverse event, Treatment-related AEs adverse events with a causality assessment of definitely related, probably related, or possibly related to the investigational product. One participant in the 15 mg cohort discontinued due to an adverse event considered unrelated to PG-102 and was included in the safety analysis set.

severely hyperglycemia setting (HbA1c > 10%), vehicle-treated animals lost ~40% of body weight over 3 months, consistent with prior reports showing comparable reductions from ~40 g at week 10 to ~25 g by week 20[19]. Even under less severe hyperglycemia (9% <HbA1c < 10%), vehicle mice only lost ~8% of body weight, highlighting that spontaneous wasting scales with baseline disease severity rather than experimental variability. Under these conditions, PG-102 uniquely decoupled glycemic control from weight loss. Unlike semaglutide or tirzepatide, which link glucose lowering with additional weight reduction, PG-102 achieved normoglycemia while substantially preserving body weight across both severe and moderate hyperglycemia models. This distinctive profile can be explained by several complementary mechanisms. First, PG-102's profound glucose-lowering likely restores normal substrate partitioning, redirecting energy utilization away from protein and lipid catabolism toward carbohydrate metabolism. By alleviating this catabolic flux—the primary driver of wasting in advanced diabetes—PG-102 helps preserve body weight despite achieving normoglycemia. Second, GLP-2 biology has been implicated in muscle preservation via the IGF-1/PI3K/Akt/FoxO3a pathway, which governs the balance between protein synthesis and degradation[37]. In a D-galactose–induced muscle atrophy model, GLP-2 significantly reversed declines in muscle weight, grip strength, and fiber cross-sectional area, while suppressing MuRF-1 and Atrogin-1 and enhancing MyoD, MyoG, and Myhc expression, effects shown to be mediated by IGF-1/PI3K/Akt/FoxO3a activation[38]. Consistent with these

findings, our preliminary in vitro data in C2C12 myotubes showed that PG-102 similarly downregulated Atrogin-1 and MuRF-1 under catabolic stress (Supplementary Fig. 11), supporting the hypothesis that dual GLP-1R/GLP-2R activation may extend GLP-2–mediated muscle preservation biology into the advanced T2D setting. Finally, modulation of the gut–muscle axis may also contribute. Gut microbiota and their metabolites are known to influence muscle mass and systemic inflammation, while GLP-2 signaling enhances intestinal barrier integrity and reduces endotoxemia[39,40]. It is therefore conceivable that PG-102, by stabilizing the gut environment, may help preserve muscle through reduced inflammatory burden and improved nutrient absorption—a hypothesis warranting further in vivo validation. Collectively, these mechanisms provide a coherent rationale for PG-102's ability to deliver glycemic control while sparing body weight—an effect particularly valuable for advanced T2D patients prone to unintentional weight loss, who may not tolerate or benefit from additional weight reduction induced by conventional GLP-1RAs.

In our phase 1 MAD study, PG-102 was well tolerated, with GI adverse events occurring at relatively low and consistent rates across doses. Nausea, one of the most common adverse effects of GLP-1RAs[4], was reported in only one-third of participants across all cohorts and showed no dose dependence. Diarrhea was similarly infrequent and comparable to placebo at therapeutic dose levels. These findings suggest that PG-102 may overcome a key limitation of current incretin therapies, which often require slow up-titration to mitigate GI

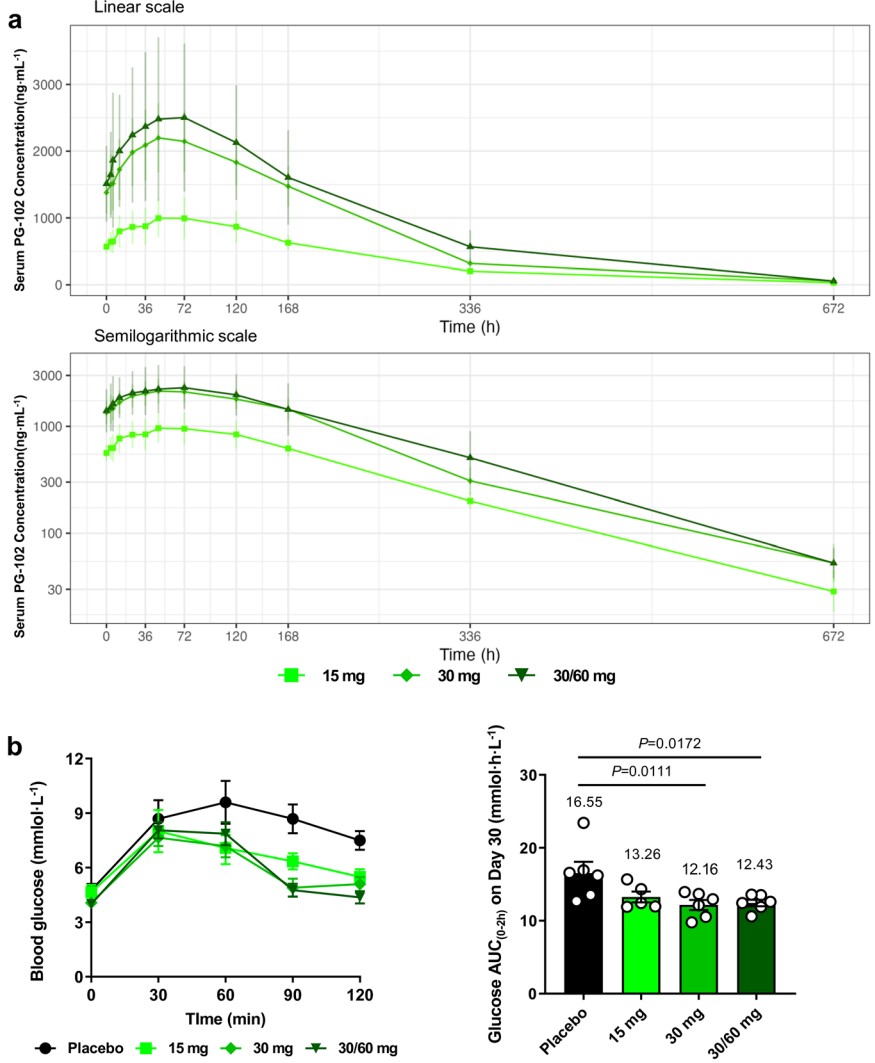

**Fig. 7 | Pharmacokinetic and pharmacodynamic profiles of PG-102 in a phase 1 multiple ascending dose study. a** Pharmacokinetic (PK) profiles of PG-102 following the final dose. Serum concentration–time curves are shown on a linear-scale (upper panel) and semilogarithmic-scale (lower panel), for each dose cohort (15 mg: $n = 5$; 30 mg: $n = 6$; 30/60 mg: $n = 6$). Data are presented as mean ± SD. **b** Pharmacodynamic (PD) assessment by oral glucose tolerance test (OGTT) performed on Day 30, 1 day after the final dose. Left, blood glucose excursion over 120 min; right, glucose area under the curve from 0 to 2 h ($AUC_{0-2h}$). Cohort sizes were placebo: $n = 6$; 15 mg: $n = 5$; 30 mg: $n = 6$; 30/60 mg: $n = 6$. Data are presented as mean ± SEM. Statistical significance was evaluated by one-way ANOVA with Dunnett's multiple comparisons test using placebo as the reference. All participants included in the PK and PD analyses completed all planned sampling time points. PK pharmacokinetics, PD pharmacodynamics, OGTT oral glucose tolerance test, AUC area under the curve, SD standard deviation, SEM standard error of the mean. Source data are provided as a Source data file.

intolerance. Several pharmacological features may underlie this favorable tolerability. First, PG-102 exhibited delayed $T_{max}$ (48.0–72.0 h), which likely blunts peak serum excursions and reduces concentration-driven GI side effects. Similar observations have been reported with other long-acting GLP-1 receptor agonists, such as dulaglutide ($T_{max}$ ~48.0 h), which does not require initial up-titration. In contrast, agents with a shorter $T_{max}$, such as semaglutide ($T_{max}$ ~24 h), typically necessitate gradual titration to mitigate GI adverse events[28], underscoring the advantage of delayed absorption in facilitating tolerability and simplifying treatment regimens. Second, PG-102's altered biodistribution pattern, likely related to its large-molecule nature, may favor peripheral actions while limiting central nervous system (CNS) exposure[41]. Exploratory biodistribution assessments with an PG-102 precursor (PG-105) suggested negligible brain penetration, consistent with expectations for Fc-based GLP-1 agonists[41]. These preliminary observations support the hypothesis that reduced CNS exposure could contribute to the favorable tolerability profile of PG-102, although direct biodistribution studies with the

current molecule remain warranted. Third, GLP-2 receptor biology is known to strengthen intestinal barrier integrity by upregulating tight junction proteins such as occludin and zonulin-1, which help prevent fluid leakage and support stool consistency[42,43]. These barrier-enhancing effects reduce endotoxin translocation and systemic inflammation, thereby contributing to both GI tolerability and metabolic stability[44]. Consistent with this, our previous work in a choline-deficient high-fat diet–fed mouse model demonstrated that PG-105 markedly upregulated the tight junction proteins occludin and zonulin-1 in both ileum and jejunum, accompanied by reductions in circulating endotoxin (LPS) levels[12]. These findings support the hypothesis that GLP-2R engagement can enhance gut barrier integrity and mitigate systemic inflammation[42,43], effects likely to be recapitulated with PG-102.

While these pharmacologic and mechanistic features provide a rationale for PG-102's favorable tolerability, several limitations and safety considerations should be noted. The present phase 1 MAD study was primarily designed to evaluate safety, tolerability, and

pharmacokinetics, and did not directly assess GLP-2R–specific endpoints such as intestinal permeability, nutrient absorption, or gut hormone secretion. Preclinical data support a contributory role for GLP-2R activation, but its precise mechanistic relevance in humans remains to be established. Regarding long-term safety, although concerns have been raised about intestinal neoplasms with chronic GLP-2 analog therapy[45,46], recent real-world evidence from a 3-year follow-up in short bowel syndrome patients showed no increase in colonic polyps, adenomas, or colorectal cancer[47]. Moreover, 26-week repeat-dose studies of PG-102 in rodents and primates revealed no evidence of intestinal or thyroid C-cell hyperplasia (Supplementary Table 1). Finally, the markedly lower GLP-2R potency of PG-102 (~3–4% relative to native GLP-2) compared with teduglutide likely minimizes the risk of intestinal trophic overstimulation. The phase 1 MAD population was predominantly male, limiting the evaluation of sex-related differences. Collectively, these findings reinforce PG-102's promising safety and tolerability profile while underscoring the need for larger, sex-balanced, and mechanistically focused studies to confirm the translational relevance of dual GLP-1/GLP-2 receptor engagement.

In summary, this study supports PG-102 as a bispecific GLP-1/GLP-2 receptor agonist that addresses a key limitation of current incretin-based therapies by dissociating glycemic control from catabolic weight loss in advanced T2D. Through coordinated effects on β-cell preservation, glucose utilization, and inflammatory modulation, PG-102 achieved sustained glycemic improvement in preclinical models characterized by severe hyperglycemia and weight loss, while maintaining body weight. In early-phase clinical evaluation, PG-102 demonstrated acceptable tolerability, a low incidence of gastrointestinal adverse events, and pharmacokinetics properties consistent with extended dosing intervals without titration. Collectively, these findings highlight the therapeutic relevance of incorporating GLP-2 biology alongside GLP-1 signaling to broaden the functional scope of incretin-based treatment strategies. Future clinical studies will be required to determine the durability of these effects and their clinical implications in patients with advanced T2D.

## Methods

### Ethical approval
All animal experiments were conducted in accordance with relevant ethical regulations and were approved by the Institutional Animal Care and Use Committee (IACUC) of GI Biome (approval number: GIB-23-02-007). All cell-based experiments were performed in compliance with institutional guidelines. The clinical study was conducted in accordance with the Declaration of Helsinki and Good Clinical Practice (GCP) guidelines, and was approved by the Ministry of Food and Drug Safety of the Republic of Korea (clinical trial authorization no. 101629) and by the Institutional Review Board of Seoul St. Mary's Hospital, the Catholic University of Korea (IRB no. KC23MDSF0619). Study progression and dose escalation were subject to review and approval by an internal Safety Review Board at prespecified stages. Written informed consent was obtained from all participants prior to study participation.

### Production of PG-102
The nucleotide sequence encoding PG-102 was synthesized (ATUM, Newark, CA, USA) and cloned into the pCGS3 expression vector. Stable cell line development was conducted by Samsung Biologics (Incheon, South Korea) using the CHOZN® ZFN-modified GS-/- cell line, which was transfected with the purified DNA construct. The transfected cells were cultured in a controlled bioreactor system, and the expressed protein was purified using a multi-step chromatographic process to ensure high purity. The final product was formulated in a designated buffer and quantified using UV spectrophotometry (Implen, CA, USA). Only batches meeting stringent predefined quality criteria, including purity, structural integrity, and biological activity, were advanced for downstream in vitro and in vivo studies.

### In vitro bioactivity assay
The bioactivity of PG-102 was evaluated using cell lines expressing GLP-1 and GLP-2 receptors, provided by Eurofins DiscoverX Ltd. (Shanghai, China) and MORE-Bio, Inc. (Hanam-si, South Korea), respectively. Cells were cultured following the manufacturers' protocols, and receptor activity assays were conducted to assess PG-102's potency. For comparison, GLP-1-NTIG Fc, GLP-2-NTIG Fc, and PG-105 analogs were prepared and tested concurrently.

### In vivo study
Twelve-week-old *db/db* mice (BKS-Leprdb/db/JOriRj, male) were obtained from Janvier-Labs (Le Genest-Saint-Isle, France) and housed under controlled environmental conditions (22–24 °C, 50–60% relative humidity, 12-h light/dark cycle) with ad libitum access to food and water. Mice were maintained on a standard laboratory chow diet (Teklad Diets 2918, Inotiv, USA; 6% kcal from fat, 44.2% kcal from carbohydrates, and 18.4% kcal from protein) throughout the study unless otherwise indicated. To model advanced-stage T2D, *db/db* mice were acclimated for two weeks to induce severe hyperglycemia, defined as baseline HbA1c > 10% and non-fasting blood glucose >500 mg/dL, prior to treatment initiation. At 14 weeks of age, mice were randomized into treatment groups (*n* = 8 per group) receiving subcutaneous injections of PG-102 (15, 30, 60, or 120 nmol/kg), semaglutide (30 nmol/kg), tirzepatide (15 nmol/kg), or vehicle every 3 days for 12 weeks. Sample sizes were guided by prior experience with the *db/db* mouse model and were designed to provide ≥80% power (α = 0.05) to detect biologically meaningful differences in metabolic outcomes. Non-fasting blood glucose and HbA1c levels were monitored at baseline and every three weeks. At the end of the study, fasting blood glucose and insulin levels were measured following a 6-h fast, and HOMA-IR and HOMA-β scores were calculated using standard formulas. Only male mice were used in this study to minimize variability associated with the estrous cycle and because male *db/db* mice exhibit more severe and consistent hyperglycemia; therefore, sex was not included as a biological variable in the study design or analysis.

### Histological and immunohistochemical analyses
At the end of the study (week 26), pancreatic tissues were harvested, fixed in 10% neutral-buffered formalin, embedded in paraffin, and sectioned at 4 μm. For histological evaluation, sections were stained with hematoxylin and eosin (H&E) to assess islet architecture. Adjacent sections were subjected to immunohistochemical staining for insulin to visualize insulin-positive islets. For immunohistochemical analysis, sections were incubated with a primary antibody against insulin (Abcam, ab181547; 1:10,000; rabbit), followed by appropriate HRP-conjugated secondary antibodies and chromogenic detection. For immunofluorescence analyses, sections were subjected to antigen retrieval, blocked, and incubated with primary antibodies against insulin (Abcam, ab181547; 1:10,000; rabbit), glucagon (Abcam, ab10988; 1:200; mouse), and Ki67 (Abcam, ab15581; 1:300; rabbit), followed by species-appropriate fluorescent secondary antibodies and nuclear counterstaining with DAPI. Images were acquired using a fluorescence microscope (ECLIPSE Ci-L; Nikon Instruments Inc., Melville, NY, USA). Quantification of positive staining areas and Ki67-positive cells was performed using ImageJ software (NIH).

### Intraperitoneal glucose tolerance test (ipGTT)
Ten-week-old C57BL/6 J mice (The Jackson Laboratory, Bar Harbor, ME, USA) were administered the indicated drugs (15 nmol/kg) and maintained for 24 h, prior to testing, including a 6-h fasting period immediately before the ipGTT. Glucose (2 g/kg body weight) was injected intraperitoneally, and blood glucose concentrations were measured at the indicated time points up to 2 h post-injection using a standard glucometer (Accu-Chek Active; Roche Diagnostics, Mannheim, Germany).

### Islet isolation and glucose-stimulated insulin secretion

Pancreatic islets were isolated through collagenase digestion of the pancreas, followed by gradient purification to obtain intact islets. Fifty size-matched islets were perifused with Krebs-Ringer bicarbonate buffer (KRB) containing 0.1% bovine serum albumin at 37 °C and 200 µL/min. After stabilization, islets were exposed to 3 mM glucose (basal) followed by 11 mM glucose supplemented with 100 nM test drugs. Perifusate fractions were collected every minute for 45 min, and insulin concentrations were quantified using an ELISA kit (ALPCO, Salem, NH, USA) to evaluate basal and glucose-stimulated insulin secretion.

### Cell viability assay

Cell viability was assessed using the MTS assay (CellTiter 96® AQueous One Solution Cell Proliferation Assay, Promega) according to the manufacturer's instructions. Briefly, INS-1 cells were seeded in 96-well plates at a density of $5 \times 10^4$ cells/well and allowed to adhere overnight at 37 °C in a humidified atmosphere containing 5% $CO_2$. Cells were then treated with the indicated compounds for 24 h following a 3-h exposure to streptozotocin (STZ). Subsequently, 20 µL of MTS reagent was added directly to each well containing 100 µL of culture medium, and the plates were incubated at 37 °C for 4 h. Absorbance was measured at 490 nm using a microplate reader (SpectraMax, Molecular Devices). Cell viability was expressed as a percentage relative to the control group (vehicle-treated cells). All experiments were performed in triplicate and independently repeated at least three times.

### In vitro glucose-stimulated insulin secretion (GSIS)

Insulin secretion was measured using the Krebs Ringer Bicarbonate (KRB) buffer (119 mM NaCl, 5 mM KCl, 2.5 mM $CaCl_2$, 1.2 mM $KH_2PO_4$, 1.2 mM $MgCl_2$, 10 mM HEPES, 25 mM $NaHCO_3$, and 0.2% BSA). Cells were pre-incubated for 2 h in the KRB buffer containing 0.2 mM glucose for glucose starvation. After pre-incubation, cells were incubated in KRB buffer containing 2- or 16.8-mM glucose (Sigma-Aldrich) at 37 °C for 30 min, respectively. Insulin released into the supernatants was measured using the Mouse Ultrasensitive Insulin ELISA kit (ALPCO) according to the manufacturer's instructions.

### Cell culture and differentiation

3T3-L1 preadipocytes and L6-GLUT4myc myoblasts were cultured in Dulbecco's Modified Eagle Medium (DMEM; Thermo Fisher Scientific, Waltham, MA, USA) and α-MEM (Thermo Fisher Scientific), respectively, supplemented with 10% fetal bovine serum (FBS) and 1% penicillin-streptomycin (P/S) at 37 °C and 5% $CO_2$. For 3T3-L1 cells, differentiation was initiated upon confluence (day 0) with DMEM containing 0.5 mM isobutylmethylxanthine (IBMX), 1 µM dexamethasone, and 10 µg/mL insulin. On day 2, the medium was replaced with DMEM supplemented with 10% FBS and 10 µg/mL insulin, refreshed every 2 days. Differentiation was completed within 6–8 days, confirmed by the presence of lipid droplets. For L6-GLUT4myc cells, differentiation was initiated at 80–90% confluence using α-MEM supplemented with 2% horse serum (HS), refreshed every 2–3 days. Differentiation into multinucleated myotubes was achieved within 5–7 days.

### Glucose uptake assay

Glucose uptake in 3T3-L1 adipocytes and L6-GLUT4myc myotubes was measured using a glucose uptake cell-based assay kit (Cayman, Ann Arbor, MI, USA), according to the manufacturer's protocol. After 2 h of serum starvation, cells were treated with the indicated drugs (300 nM) and insulin simultaneously for 10 min. Cells were washed three times with Krebs buffer (20 mM HEPES, 5 mM $KH_2PO_4$, 1 mM $MgSO_4$, 1 mM $CaCl_2$, 136 mM NaCl, 4.7 mM KCl, pH 7.4), then incubated with 500 µg/ml 2-NBDG, a fluorescence-labeled deoxyglucose analog, for 20 min at room temperature. Reactions were terminated by washing with ice-cold Krebs buffer. Cells were lysed using Mammalian Protein Extraction Reagent Thermo Fisher Scientific) and centrifuged at $15,000 \times g$ for 10 s. After removing the supernatant, assay buffer added, and 2-NBDG fluorescence was measured using a microplate reader (excitation/emission = 485/535 nm). Glucose uptake was normalized to protein content quantified using a bicinchoninic acid (BCA) assay (Thermo Fisher Scientific).

### Nonclinical toxicology studies

GLP-compliant repeat-dose toxicity studies were conducted in cynomolgus monkeys (Macaca fascicularis; males 2–4 years, females 3–5 years) at an AAALAC-accredited facility (Ina Research Inc., Japan) in accordance with OECD GLP principles and 21 CFR Part 58. All procedures were approved by the Institutional Animal Care and Use Committee (IACUC; protocols YG21113 and ZO23242) and performed in compliance with applicable animal welfare regulations.

Two repeat-dose studies were performed: a 4-week study with a 4-week recovery phase and a 26-week study. In both studies, animals were randomized by sex to receive vehicle (formulation buffer) or PG-102 and were administered subcutaneously once weekly.

In the 4-week study, PG-102 was administered at 2, 6, or 20 mg/kg ($n = 5$/sex for vehicle and 20 mg/kg; $n = 3$/sex for 2 and 6 mg/kg). In the 26-week study, PG-102 was administered at 2, 5, or 10 mg/kg once weekly for 26 consecutive weeks.

Animals were housed individually under controlled environmental conditions (22 ± 2 °C; 40–70% humidity; 12-h light/dark cycle) with environmental enrichment and ad libitum access to standard primate diet and water.

Animals were monitored at least twice daily throughout the studies, including pre- and post-dose observations on dosing days. Clinical evaluations included general condition, posture, activity, food consumption, gastrointestinal output, and body weight. Humane endpoints were prospectively defined prior to study initiation and included sustained anorexia, marked body weight loss, persistent gastrointestinal disturbances, or other signs of clinical deterioration compromising animal welfare. No animals met predefined humane endpoint criteria requiring unscheduled euthanasia in either study.

At scheduled termination, animals were anesthetized and euthanized under deep anesthesia in accordance with accepted veterinary practice. Complete gross necropsy was performed, and organs and tissues were collected, fixed in neutral-buffered formalin, processed routinely, stained with hematoxylin and eosin, and examined by light microscopy.

Parallel 26-week GLP repeat-dose studies in Sprague–Dawley rats were conducted under comparable conditions and are described in the Supplementary Methods.

### Phase 1 study design; Pharmacokinetic analysis, safety, and pharmacodynamic evaluation

A randomized, double-blind, multiple ascending dose phase 1 study was conducted at a single clinical site in the Republic of Korea (Seoul St. Mary's Hospital, The Catholic University of Korea) to evaluate the safety, tolerability, and pharmacokinetics of subcutaneous PG-102 in healthy individuals with overweight (body mass index 25–30 kg/m²), with exploratory pharmacodynamic and biomarker assessments.

The reported study corresponds to Part B of a three-part Phase 1 clinical trial. The first participant was enrolled on March 4, 2024, and the last participant completed enrollment on July 15, 2024.

Progression between dose levels was prespecified and permitted only after safety review and approval by an internal Safety Review Board at each stage. The trial was preregistered at ClinicalTrials.gov (NCT06309667) on February 14, 2024, and conducted in accordance with the Declaration of Helsinki and the International Council for Harmonisation Good Clinical Practice guidelines. The study is reported

in accordance with CONSORT 2010, and a completed checklist is provided in the Supplementary Information (Supplementary Note 1).

The full clinical study protocol and prespecified statistical analysis plan are available in the Supplementary Information (Supplementary Notes 2 and 3).

The primary outcomes were safety and tolerability, assessed by the incidence, severity, and causality of adverse events and by changes in clinical laboratory parameters and physical examinations. Secondary outcomes included the pharmacokinetic profile of PG-102, assessed using serum concentration–time data and non-compartmental analyses, with exploratory pharmacodynamic and biomarker endpoints. Safety assessments were conducted from baseline through end-of-study follow-up, and pharmacokinetic and exploratory assessments were performed at prespecified time points, including serial sampling up to 672 h after the final dose.

As is typical for early-phase clinical trials, no formal statistical hypothesis testing or sample size calculation was performed. Sample size was determined based on practical considerations, prior experience with similar phase 1 study designs, and the objective of obtaining sufficient safety and pharmacokinetic data to support dose escalation decisions. Continuous variables were summarized descriptively (n, mean, standard deviation, median, minimum, maximum), and categorical variables by frequency and percentage.

Sex and gender were considered in the study design without restrictions on enrollment. All participants self-reported their sex at screening. Of 67 individuals screened, 31 met eligibility criteria, and 24 were randomized. Enrollment was defined as randomization, with participants assigned within each cohort in a 6:2 ratio to receive PG-102 or a placebo. Randomization was performed using a predefined allocation schedule prepared by the study sponsor. Allocation concealment was maintained through centralized assignment procedures to ensure blinding. Participants, investigators, and study staff involved in clinical conduct and outcome assessment were blinded to treatment allocation. PG-102 and placebo were identical in appearance and administered in the same manner to maintain blinding. Cohorts were as follows: Cohort 1 (15 mg weekly), Cohort 2 (30 mg weekly), and Cohort 3 (escalating 30/30/60/60/60 mg weekly). This allocation resulted in 18 participants receiving PG-102 and 6 receiving a placebo. Among the 24 randomized participants, 22 (91.7%) were male, and 2 (8.3%) were female (Table 1). Given the very limited number of female participants, sex- and gender-based analyses were not performed, as they would not provide meaningful conclusions.

In the 15 mg cohort, one participant discontinued after the fourth dose due to an adverse event considered unrelated to the investigational product. This participant subsequently withdrew but completed the scheduled follow-up visit, including all safety assessments. Accordingly, the participant was included in the safety analysis set but excluded from the efficacy and PK analysis sets.

Participants received 5 weekly doses of PG-102 or a placebo and were followed for two weeks after the last dose. Serial blood samples were collected for PK analysis at pre-specified intervals up to 672 h after the final dose (pre-dose, 4, 6, 12, 24, 36, 48, 72, 120, 168, 336, and end-of-study at 672 h). Serum concentrations of PG-102 were quantified using a validated immunoassay, and non-compartmental analysis was performed to determine PK parameters. Safety and tolerability were assessed throughout the study by monitoring adverse events (AEs), clinical laboratory tests, and physical examinations. AEs were evaluated for severity and causality, and classified as not related, unlikely, possible, probable, or definite. "Treatment-related" AEs were defined as those assessed as definite, probable, or possible.

### Statistical analysis

Data are expressed as mean ± standard error of the mean (SEM), unless stated otherwise. Statistical analyses were performed using GraphPad Prism (version 9.0, GraphPad Software, San Diego, CA, USA). For comparisons among multiple groups, one-way ANOVA followed by Tukey's post hoc test was applied to assess all possible pairwise differences. For planned comparisons against a single treatment of interest, Dunnett's test was applied using the relevant reference group (PG-102 for preclinical experiments, placebo for clinical datasets). Quantitative histological data were normalized to tissue area or cell count using ImageJ (NIH, Bethesda, MD, USA). In the clinical dataset, no formal correction across multiple PD endpoints was applied, and results should be interpreted as exploratory. Statistical significance thresholds were $*p < 0.05$, $**p < 0.01$, and $***p < 0.001$.

### Reporting summary

Further information on research design is available in the Nature Portfolio Reporting Summary linked to this article.

## Data availability

All data generated in this study are provided in the Supplementary Information and the Source Data files published alongside this article. The Source Data files contain the minimum dataset required to interpret, verify, and extend the findings reported in this study. All clinical and preclinical data underlying the figures and tables are included in the Source Data files, in accordance with Nature Portfolio data availability policies. The clinical study protocol and the statistical analysis plan are provided in the Supplementary Information as Supplementary Notes 2 and 3, respectively, and are referenced in this Data Availability Statement. No additional datasets were deposited in external public repositories. This study did not generate or analyze custom code. Source data are provided with this paper.

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

## Acknowledgements

This research was supported by Korea Drug Development Fund funded by Ministry of Science and ICT, Ministry of Trade, Industry, and Energy, and Ministry of Health and Welfare (RS-2024-00335179, Republic of Korea; awarded to S.Y., S.W.K., K.S., S.L., and J.K.). This study was sponsored by ProGen Co., Ltd., which was involved in the study design, data collection, data analysis, and manuscript preparation. We thank Dr. Kun-Ho Yoon (ProGen Co., Ltd.) for his insightful review of the manuscript. We are also grateful to Dr. Seung-Hoon Han and Dr. Sung-Pil Han (The Catholic University of Korea, Seoul St. Mary's Hospital) for conducting the clinical trial. All authors had full access to the study data, contributed to data interpretation, and approved the final version of the manuscript. No medical writing assistance was used.

## Author contributions

Y.C.S. and S.Y. conceived and designed the study. J.R. conducted the experiments. S.Y., S.W.K., K.S., S.L., and J.K. analyzed the data, prepared

the figures and tables, and drafted the manuscript. Y.C.S., S.Y., and S.W.K. critically reviewed the study design, interpreted the results, and revised the manuscript. Y.C.S. had full access to all data and is responsible for its integrity and the accuracy of the analyses.

## Competing interests

Y.C.S. and S.Y. are inventors on patents related to this work (PCT/KR2021/003127 & PCT/KR2021/003128). Y.C.S. is a current employee of SL BiGen, Inc. S.I. is a former employee of SL BiGen, Inc. and now a current employee of ProGen Co., Ltd. J.R. is a former employee of both SL BiGen, Inc. and ProGen Co., Ltd. S.W.K., K.S., S.L., and J.K. are current employees of ProGen Co., Ltd.

## Additional information

Sang-In Yang ⓘ [1,2,4], Sae Won Kim ⓘ [1,4], Kyung-Hwa Son ⓘ [1], Seung-Ah Lee ⓘ [1], Jong-Gyun Kim[1], Jae-Il Roh[1,2] & Young Chul Sung ⓘ [2,3] ✉

[1]ProGen Co. Ltd., Seoul, Republic of Korea. [2]SL BiGen Inc., Incheon, Republic of Korea. [3]Division of Integrative Biosciences and Biotechnology, Pohang University of Science and Technology (POSTECH), Pohang, Republic of Korea. [4]These authors contributed equally: Sang-In Yang, Sae Won Kim. ✉e-mail: ycsung@postech.ac.kr

