## [Transparent Peer Review file · Nature Communications]

Bispecific GLP-1/GLP-2 Agonism in Advanced Type 2 Diabetes: Preclinical Characterization and a Randomized, Double-blind, Placebo-controlled Phase I Trial

Corresponding Author: Professor Young Chul Sung

Version 0:

Reviewer comments:

Reviewer #1

(Remarks to the Author)

This manuscript introduces PG-102, which targets both GLP-1 and GLP-2 receptors to treat type 2 diabetes. The manuscript presents results from in vitro studies evaluating the properties of PG-102, mouse studies comparing the effects of PG-102 with semaglutide and tirzepatide, and a phase 1 multiple ascending dose study in humans assessing the safety and pharmacokinetic and pharmacodynamic profiles of PG-102. Overall, the approach of the described studies seems reasonable. My primary concern is that parts of the manuscript need further clarification, including some inconsistencies in different parts of the manuscript. My specific comments are provided below:

1. In the description of the results in the text: There were a couple places where extra care should be taken to avoid over-interpreting some of the small numerical effects observed. In particular:
 - a. On line 198, when describing the comparative efficacy and mechanistic evaluation of PG-102 vs. semaglutide and tirzepatide: The text state that “semaglutide and tirzepatide treatment partially improved islet structure and beta-cell area, but restoration was limited.” Given that the islet area and beta-cell area both look nearly identical between the vehicle group and tirzepatide group based on Figures 3g and 3h, I would be hesitant to comment that tirzepatide improved these two outcomes.
 - b. In the description of the safety results from the phase 1 multiple ascending dose study (line 280), it states “vomiting ... demonstrated dose-dependent increases...” For vomiting, there were a very small number of events in all treatment groups (ranged from 0 to 2 events), with 0 events in both the placebo group and the 15 mg PG-102 group. Therefore, I would be hesitant to emphasize the dose-dependent relationship for vomiting too strongly, due to concern that these results may possibly be due to chance.
2. In the description of the phase 1 multiple ascending dose study in the results section, on line 266, the reference to “Extended Data Table 1” should be changed to “Extended Data Table 2” (subject disposition across cohorts).
3. There were a couple sets of previously unpublished results that were referenced briefly in the manuscript, but not described in detail in the methods. In particular:
 - a. The Discussion section (lines 334-337) refers to results from administering PG-102 to two naturally occurring diabetic feline patients, but these procedures were not described in the methods, nor were the results described prior to the Discussion section.
 - b. In Extended Data Table 4, data are presented from a GLP toxicity study in monkeys, but these procedures were not described in the methods, nor were the results described in the manuscript.
4. For the in vivo study comparing PG-102 to semaglutide and tirzepatide in mice, the sample size should be provided in the methods section.
5. Also for the in vivo study comparing PG-102 to semaglutide and tirzepatide in mice, the age of the mice is inconsistent in the description of the methods (12-weeks) vs. in the figure legends (14-weeks). Please confirm the correct age of the mice,

and ensure that this is correct throughout the manuscript.

6. In the description of the phase 1 study design and PK analysis in the methods section, on lines 585-586, it states “Serial blood samples were collected for PK analysis at pre-specified intervals up to 696 hours after the final dose.” The “pre-specified intervals” should be described in the text.

7. In the description of the statistical analysis in the methods section (line 592), it states “one-way ANOVA with Tukey’s post hoc test was used for multi-group comparisons...” Please specify whether all pairs of groups were tested using Tukey’s post hoc test, or if only a subset of tests of interest were assessed (and if so, which subset of tests were assessed). This will be helpful for interpreting whether lack of an indication of statistical significance in the results section and figures indicates a non-statistically significant test result or simply that the comparison was not tested. In addition, the figure legends throughout the manuscript state that statistical significance was assessed using one-way ANOVA, but some of the figure legends do not state the use of Tukey’s post hoc test; the use of Tukey’s post hoc test should be indicated in the figure legends where applicable.

8. In the figures (e.g., Figure 2), it would be helpful to state the timeframe for cross-sectional results in the figure legend. For example, the legend for Figure Panels 2c and 2d-g state that the results are from end-of-study, but it would be helpful to re-state the timeframe for end-of-study in the legend for easy reference.

9. Figure 6a includes two plots stacked on top of each other. Based on the figure legend, the top plot is on the linear-scale, and the bottom plot presents the same data on the semilogarithmic-scale. It would be helpful to include headings on these two plots to make it clear what is the difference between the two plots.

10. Table 1: It is not clear to me what is meant by “treatment-related” adverse events. Please provide a definition for this category of adverse events in the manuscript text and in the description for Table 1.

11. Was Extended Data Table 1 (Thyroid C cell and gastrointestinal hyperplasia in long-term repeated dose toxicity studies in rats and cynomolgus monkeys) referred to anywhere in the manuscript text?

12. I am a bit confused about screening and enrollment for the phase 1 multiple ascending dose study, based on the data presented in Extended Data Table 2. In particular:

a. It would be helpful to specify in the table heading and/or description that the numbers presented here are for the phase 1 multiple ascending dose study.

b. There is no column for the placebo group provided in this table. The relevant information for the placebo group should be added.

c. The number of volunteers screened and number of volunteers not enrolled are provided in the table by treatment group. Were participants randomized before enrolling in the study? If so, what was the justification for this?

d. The numbers of subjects enrolled in this table do not seem to match the sample size provided in the methods section text, or in the other tables related to the phase 1 multiple ascending dose study. This table states that there were 8 people enrolled in each of the 3 PG-102 dose cohorts. However, the methods section text states that there were a total of 24 participants, randomized in a 6:2 ratio for the 3 PG-102 dose cohorts and placebo, implying that there should be 6 participants in each treatment group, including each of the PG-102 dose cohorts. The other tables and figures (e.g., Table 1, Extended Data Table 3, Extended Data Figure 2) also imply a total sample size of 24, with 6 participants in each of the 3 PG-102 dose cohorts and in the placebo group. Please confirm that the sample sizes are correct throughout the manuscript.

e. Similar to the above point, in Extended Data Figure 2, the numbers who were screened, not eligible, and met inclusion/exclusion criteria are consistent with the total numbers across the 3 PG-102 dose cohorts from Extended Data Table 2 (i.e., excluding the placebo group). Should the numbers in Extended Data Figure 2 be revised to include the corresponding numbers from the placebo group, or were the numbers tabulated incorrectly in Extended Data Table 2? Please revise as needed to ensure that the numbers presented in Extended Data Table 2 and Extended Data Figure 2 are both correct.

f. Under “Reason for screening failure”, there were a total of 6 participants who failed screening for “other” reasons. Are these the same 6 participants who were excluded from the study due to exceeding the subject limit, according to Extended Data Figure 2? If the reason for screening failure is the same for all 6 participants in the “other” category (i.e., due to exceeding the subject limit), then this reason can be stated in the table, instead of labeling the row as “others”.

13. For the phase 1 multiple ascending dose study, based on Extended Data Table 2, there was one participant who dropped out of the study. Did this participant drop out of the study completely (i.e., stopped participating in data collection procedures), or did they just discontinue the randomized medication? Was the participant who dropped out included in any of the analyses, and if so which ones? Please specify this information in the description of the phase 1 multiple ascending dose study in the methods section.

14. Extended Data Table 5: There is an asterisk next to T_{max} in the table, but the meaning of the asterisk is not defined anywhere in the table or legend.

15. Extended Data Table 6: On line 917 of the figure legend, what is the meaning for the numbers in parentheses following exendin (9-39) and GLP-2 (3-33)?

16. Extended Data Figure 2: In the second box on the right side, the word “consent” is included twice.

17. Extended Data Figure 3:

- a. Why was semaglutide excluded from this analysis of the moderate and mild hyperglycemia mouse models (even though it was included in the analysis of the severe hyperglycemia mouse model in Figure 3)? This should be explained in the figure legend and/or in the manuscript text.
- b. Panels d-f (results from the mild hyperglycemia mouse model) appear to be missing.
- c. The dose for tirzepatide appears to be incorrect in the legend indicating the colors and symbols for the different treatment groups (it is listed as 30 nmol/kg, but should be 15 nmol/kg).

18. Extended Data Figure 4: The procedures for this comparison of PG-105 and PG-102 do not appear to be described in the methods section.

Reviewer #2

(Remarks to the Author)

Summary

This manuscript presents the preclinical and early clinical development of PG-102, a bispecific GLP-1/GLP-2 receptor agonist, for the treatment of type 2 diabetes (T2D). The authors demonstrate PG-102 in vitro characterization and in vivo efficacy in preclinical models and report favorable tolerability and pharmacokinetics in a Phase 1 multiple ascending dose (MAD) study. The central claim is that PG-102 achieves glycemic control without inducing weight loss, distinguishing it from existing GLP-1-based therapies.

While the manuscript is ambitious and addresses an important clinical need, several key mechanistic claims are insufficiently supported. In particular, the dual pharmacology of PG-102 is not convincingly demonstrated in vivo. Additionally, some experimental designs and interpretations require clarification or further validation.

Major Comments

1. The manuscript should include a direct in vivo comparison between PG-102, the GLP-1-NTIG Fc, the GLP-2-NTIG Fc, and a loose combination of the GLP-1 and GLP-2 NTIG Fc's to validate the synergistic effects of dual receptor targeting, especially for glucose control and islet protection.
 - a. PG-102 exhibits only ~3.4% of native GLP-2R potency. It is unclear whether this level of activity is sufficient to elicit meaningful in vivo effects. A comparison with a GLP-1-only antibody is essential to isolate the contribution of GLP-2R agonism.
 - b. While the adipocyte data support dual receptor synergy, similar experiments in islets or in vivo are lacking. Without these, the role of GLP-2 on the central pharmacologic action of PG-102 remains speculative.
2. The manuscript claims enhanced tolerability in humans, but it is unclear what this is relative to. No marketed comparators were included in the Phase 1 study. Please clarify the basis for this claim.
3. The authors should report whether PG-102 activates other class B GPCRs such as GCGR or GIPR. This is particularly important given the structural similarity and potential for off-target effects.
4. The manuscript uses tirzepatide as a comparator in rodents, but its GIPR activity is limited in this species. This limitation should be acknowledged and discussed.
5. The reported 40% spontaneous weight loss in vehicle-treated db/db mice is unusually high. The authors should provide citations or additional data to support this observation. The supplementary data show only ~7% weight loss in a milder glucose intolerance model, which raises concerns about reproducibility.
6. The authors mention reduced CNS exposure as a potential mechanism for improved tolerability. Have they performed ICV injections or biodistribution studies to support this?
7. The claim that PG-102 enhances insulin secretion over semaglutide and tirzepatide is not statistically significant (line 215). Similarly, MafA expression is not significantly upregulated. These points should be revised or qualified.
8. The manuscript claims anti-inflammatory effects of PG-102 (line 359), but no supporting data are shown. These claims should be either substantiated or removed.
9. Plot the AUC for glucose and drug exposure (Figure 6a vs Figure 6b) to show the relationship between exposure and response.

Minor Comments

Line 84: Explicitly state the goal of achieving glycemic control without weight loss.

Line 109: Clarify that no marketed comparators were included in the Phase 1 study.

Line 218: Cite El, Douros et al. Nature Metabolism 2023 to support the role of GIPR in the pharmacology of tirzepatide.

Line 324: Replace "potent" with "effective" to avoid overstatement.

Lines 336–337: Provide data or citations for feline case studies.

Lines 348–354: Add citations to support mechanistic claims.
Line 358: Specify which other cell types co-express GLP-1R and GLP-2R.
Line 367–370: Reframe speculative mechanisms as hypotheses rather than conclusions.
Line 388: Show the data referenced.
Line 431: If available, include biodistribution data for PG-102 and dulaglutide.

Reviewer #3

(Remarks to the Author)
Strengths

Innovative dual-agonist design with translational potential

The study introduces a well-engineered GLP-1/GLP-2 dual-agonist supported by a mechanistically sound rationale. The combination of GLP-1R-mediated glycaemic control and GLP-2R-driven intestinal and systemic homeostasis offers a potential therapeutic niche in advanced type 2 diabetes, cachexia, or malabsorption.

Comprehensive early-phase study design

The manuscript captures a broad spectrum of tolerability, exposure, and PD signals in healthy adults. The inclusion of multiple ascending dose arms and detailed PK/PD endpoints (including DEXA, OGTT, HbA1c, FMI/LMI) supports an informative pharmacological characterisation. Albeit, the vast majority of these variables are not presented in the current manuscript.

Preclinical comparator data strengthen translational appeal

Head-to-head comparisons with semaglutide and tirzepatide in db/db mice highlight superior glucose-lowering efficacy independent of body weight, suggest a favourable pharmacodynamic dissociation.

Weakness

No formal statistical hypothesis or power justification provided

Although Phase 1 trials are typically exploratory, the manuscript would benefit from explicit clarification regarding sample size rationale and statistical assumptions for key PD endpoints (e.g., body weight change, HbA1c).

Limited discussion of GLP-2R-specific effects in humans (efficacy and safety)

While the dual-agonist concept is central, the manuscript provides little evidence or speculation about the contribution of GLP-2R agonism per se. For instance, GI permeability markers, gut hormones, or nutrient absorption were not assessed, weakening the mechanistic interpretation of the GLP-2R arm. Especially, the safety concerns related to long-term GLP-2 RAs in humans linked to intestinal neoplasms needs to be addressed.

PD effects in humans appear modest

Only glycaemic effects of PG-102 treatment are presented in this manuscript which is insufficient to thoroughly assess the effects of the compound. Body weight, body composition and hsCRP data needs to be presented as well. But most importantly, the effect of PG-102 appears fairly modest compared to placebo treatment. This may reflect high baseline insulin sensitivity, insufficient duration, or limited GLP-1R potency at early dose levels. The glucometabolic effect of GLP-2 receptor activation in humans remain controversial.

Overall level of enthusiasm

This is a well-conceived and well-executed early-phase translational study introducing a novel dual GLP-1/GLP-2 receptor agonist delivering exciting glycaemic results independent of change in body weight in rodents. Although the human PD effects are modest and the absence supporting data (insulin, Matsuda-index, change in body weight, body composition and inflammatory cytokines) are disappointing. The preclinical results (particularly in the db/db model) are compelling. The manuscript will be of interest to the field of incretin biology and metabolic pharmacology, but it is very heavy on the preclinical which is a disappointment. I recommend publication pending revision and the addition of more human data.

Additional comments

Below are some suggestions aiming to improve the manuscript.

- In general

- o Line 319: In T2D, metformin is the first-line treatment for hyperglycemia. Please correct the sentence
- o Line 325: Please provide insight into how many participants in the STEP-2 or STEP-6 and SURPASS trials obtain 'treat-to-target' success with an HbA1c below 5.7%
- o Line: 348-353: Please provide references supporting these claims.
- o Line 359: The author has not provided data supporting this claim of anti-inflammatory effects of PG-102
- o Line 370: Please update this section with a remark alluding to the anti-inflammatory effects observed in the present study. If, please inform the reader, that the anti-inflammatory effects of PG-102 otherwise are purely speculative
- o Line 371-374: Please provide reference
- o Line 590: Clarify whether any correction for multiple testing was applied across PD endpoints.
- o Line 596: Please add supporting information to the sample size calculation

- o Line 809 (Fig. 6): Please provide more details with regards to exact N per timepoint if sample size varies
- o Line 924 (Extended Data Fig. 2.): please add reason for dropout.

Version 1:

Reviewer comments:

Reviewer #1

(Remarks to the Author)

This is a revised manuscript that evaluates PG-102, which targets both GLP-1 and GLP-2 receptors to treat type 2 diabetes, based on in vitro studies, mouse studies comparing to semaglutide and tirzepatide, and a phase 1 multiple ascending dose study in humans. The revised manuscript is much clearer and improved, and in my opinion it is nearly ready for publication. There are a few minor remaining issues that should be addressed, which are described below:

1. In the Results, "Phase 1 MAD trial of PG-102: safety, PK and PD evaluation" section, on line 273: I would suggest clarifying that the 6:2 randomization ratio is for PG-102:placebo. For example, the sentence could be revised as "... randomized in a 6:2 ratio to receive PG-102 or placebo within three dose cohorts..."
2. In the Results, "Phase 1 MAD trial of PG-102: safety, PK and PD evaluation" section, on lines 283-284: Since the AEs can be classified as treatment-related, not the participants, I suggest re-wording the last part of this sentence as "...with AEs considered to be treatment-related in 18 participants (75.0%; Table 1)."
3. In the Methods, "Phase 1 study design; PK analysis, safety and PD evaluation" section, lines 560-561: As this sentence is currently written, it is a bit confusing since it refers to the three cohorts from the MAD study without providing information defining the cohorts. Most of the information from this sentence is clearly included later in this section on lines 570 – 572, and so the sentence from lines 560 – 561 could be deleted.
4. Please define all abbreviations from the figures and tables in the corresponding figure legend or table footnote, including SEM, SD, PK, PD, MAD, AUC.
5. The last sentence in the legend for Figure 6 states "All randomized participants completed every planned PK and PD sampling time point." However, based on the Methods (lines 579-580) and the sample sizes listed in the figure legend, it seems that this statement is not quite accurate, since 1 randomized participant in the 15 mg group who discontinued treatment was excluded from the PK/PD analyses. Instead, this sentence could be re-written as "All participants included in the analyses completed every planned PK and PD sampling time point."
6. In Table 1, the number of participants with discontinuation due to AEs for the 15 mg PG-102 is listed as 0. Is this correct? Based on the Methods (lines 579-580), it seems that the 15 mg group had 1 discontinuation due to AEs (but there were 0 discontinuations due to treatment-related AEs).
7. In the Supplementary Methods, "In vivo study" section, lines 35-36: This sentence states that the mice were "allocated by baseline glucose, HbA1c, and body weight into three groups." Does this mean that treatment group allocation was designed to achieve balance of baseline glucose, HbA1c, and body weight across groups (e.g., by "matching" mice based on these factors)? If so, I suggest that the following re-wording of this sentence would more clearly explain how this allocation was done: "...allocated into three groups (vehicle, tirzepatide, PG-102; n=6 per group), ensuring between-group balance on baseline glucose, HbA1c, and body weight."
8. In the Clinical Supplementary Results, "Exploratory exposure-response relationship between PK exposure and OGTT glucose excursion" section: Be careful not to overstate the relationship identified between PK exposure and OGTT glucose excursion. The R², Pearson r, and Spearman ρ are all small in magnitude, and so I would not have much confidence about whether the negative relationship identified represents a real biological relationship vs. is due to random error.
9. In Extended Data Table 5: Note that the notation $AUC_{[0-\infty]}$ is used in the table, but the notation $AUC_{[0-\infty]}$ is used in the table description. Please revise so that the notation is consistent between the table and the description.

Reviewer #2

(Remarks to the Author)

The authors have adequately addressed my comments and I feel the manuscript is appropriate for publication in this form.

Reviewer #4

(Remarks to the Author)

Reviewer Comments on the Revised Manuscript

The revised manuscript comprehensively addresses the original comments from Reviewer 3 and strengthens the scientific rigor of the work. The integration of preclinical and clinical data supporting PG-102 as a bispecific GLP-1/GLP-2 receptor agonist for advanced type 2 diabetes (T2D) is compelling, particularly its ability to decouple glycemic control from weight loss. However, 1 key area require further refinement to enhance mechanistic clarity, generalizability, and translational relevance. Below are specific suggestions for revision:

Mechanistic Link Between Low GLP-2R Potency and In Vivo Efficacy

PG-102 exhibits only ~3.4% of native GLP-2R potency in vitro (Fig. 1c) but delivers meaningful in vivo effects (e.g., enhanced peripheral glucose uptake, β -cell protection). While the manuscript attributes this discrepancy to "cis-co-engagement" of GLP-1R/GLP-2R, direct evidence linking this receptor trafficking phenotype to downstream GLP-2R-mediated signaling in human cells/tissues is lacking.

Add a discussion section explicitly addressing how cis-co-engagement compensates for low GLP-2R potency, citing relevant literature on receptor trafficking and signaling crosstalk.

In Conclusion, The manuscript presents a novel and clinically relevant therapeutic candidate for advanced T2D. Addressing the above points will enhance mechanistic rigor, generalizability, and translational clarity, strengthening the manuscript's impact in the field of incretin biology and metabolic pharmacology. With above revisions, the work will be well-positioned for publication and future clinical development.

Reviewer #1

This manuscript introduces PG-102, which targets both GLP-1 and GLP-2 receptors to treat type 2 diabetes. The manuscript presents results from in vitro studies evaluating the properties of PG-102, mouse studies comparing the effects of PG-102 with semaglutide and tirzepatide, and a phase 1 multiple ascending dose study in humans assessing the safety and pharmacokinetic and pharmacodynamic profiles of PG-102. Overall, the approach of the described studies seems reasonable. My primary concern is that parts of the manuscript need further clarification, including some inconsistencies in different parts of the manuscript. My specific comments are provided below:

Comment #1

In the description of the results in the text: There were a couple places where extra care should be taken to avoid over-interpreting some of the small numerical effects observed. In particular:

a. On line 198, when describing the comparative efficacy and mechanistic evaluation of PG-102 vs. semaglutide and tirzepatide: The text state that “semaglutide and tirzepatide treatment partially improved islet structure and beta-cell area, but restoration was limited.” Given that the islet area and beta-cell area both look nearly identical between the vehicle group and tirzepatide group based on Figures 3g and 3h, I would be hesitant to comment that tirzepatide improved these two outcomes.

Answer #1a

As suggested by the reviewer, we have revised the **Results** section (Line 177 of the revised manuscript) to clarify that tirzepatide did not produce measurable improvements in islet or β -cell area compared with the vehicle group. In addition, we have refined the surrounding paragraph (Lines 175–183) for greater clarity and consistency with the data shown in Figures 3d-3j, while maintaining a professional and cohesive tone throughout the manuscript.

b. In the description of the safety results from the phase 1 multiple ascending dose study (line 280), it states “vomiting ... demonstrated dose-dependent increases...” For vomiting, there were a very small number of events in all treatment groups (ranged from 0 to 2 events), with 0 events in both the placebo group and the 15 mg PG-102 group. Therefore, I would be hesitant to emphasize the dose-dependent relationship for vomiting too strongly, due to concern that these results may possibly be due to chance.

Answer 1b

As suggested by the reviewer, we have removed the dose-dependent interpretation for vomiting and revised the **Results** section (Line 287–288 of the revised manuscript) to state that vomiting was infrequent overall and absent in both the placebo and 15 mg PG-102 groups, without implying a dose relationship. In addition, we have refined the subsequent sentences (Lines 287–291) to improve clarity and ensure consistency in the description of other gastrointestinal events within the same paragraph.

Comment #2

In the description of the phase 1 multiple ascending dose study in the results section, on line 266, the reference to “Extended Data Table 1” should be changed to “Extended Data Table 2” (subject disposition across cohorts).

Answer #2

As suggested by the reviewer, we have corrected the **Results** section reference from ‘**Extended Data Table 1**’ to ‘**Extended Data Table 2**’ (Line 276 of the revised manuscript)

Comment #3

There were a couple sets of previously unpublished results that were referenced briefly in the manuscript, but not described in detail in the methods. In particular:

a. The Discussion section (lines 334-337) refers to results from administering PG-102 to two naturally occurring diabetic feline patients, but these procedures were not described in the methods, nor were the results described prior to the Discussion section.

Answer #3a

The exploratory statement regarding naturally diabetic feline patients (Lines 334–337 of the original manuscript) has been removed from the revised manuscript, because of the following reasons.

The observations were derived from pilot cases conducted at a collaborating veterinary hospital in Korea, directly performed by the attending veterinarian with verbal consent obtained from pet owners. At the time of the initial manuscript preparation, our understanding was that specific regulatory frameworks for clinical experimentation in companion animals in Korea were not yet established, and such exploratory cases could be reported descriptively.

However, upon further review during the revision process, we recognized that U.S. and international standards require companion-animal studies to adhere to ethical guidelines largely analogous to those for human clinical research, including formal ethical review and documented owner consent.

Accordingly, we have excluded all feline exploratory statements from the revised manuscript. For transparency in this reviewer response only, we provide a brief descriptive summary of the observations (**Figure 1**), noting that these were not designed or powered as controlled experiments and therefore are not presented as validated preclinical data. The present manuscript now focuses strictly on validated preclinical models and the controlled human Phase 1 study, which we believe best upholds ethical integrity while ensuring scientific rigor.

Figure 1. Exploratory glucose-lowering effects of PG-102 in naturally diabetic cats. Two diabetic cats (Cat A: domestic shorthair, 16.8 years, obese; Cat B: Russian Blue, 1.9 years, underweight) were treated with PG-102 (0.3 mg/kg, subcutaneous, once weekly for 4 weeks). Fasting blood glucose, body weight, and serum cholesterol were monitored during follow-up.

b. In Extended Data Table 4, data are presented from a GLP toxicity study in monkeys, but these procedures were not described in the methods, nor were the results described in the manuscript.

Answer #3b

As suggested by the reviewer, a description of the GLP toxicity study procedures has been added to the **Supplementary Methods**, and the corresponding results have been clarified in **Extended Data Table 4** (Lines 278–281 of the revised manuscript).

Comment #4

For the in vivo study comparing PG-102 to semaglutide and tirzepatide in mice, the sample size should be provided in the methods section.

Answer #4

As suggested by the reviewer, we have added the sample size for the in vivo study to the **Methods** section (Line 485 of the revised manuscript), specifying that each treatment group consisted of $n = 8$ *db/db* mice

Comment #5.

Also, for the in vivo study comparing PG-102 to semaglutide and tirzepatide in mice, the age of the mice is inconsistent in the description of the methods (12-weeks) vs. in the figure legends (14-weeks). Please confirm the correct age of the mice, and ensure that this is correct throughout the manuscript.

Answer #5

As suggested by the reviewer, we confirm that all in vivo experiments were conducted in 14-week-old *db/db* mice at treatment initiation. The **Methods** section has been revised to clarify that mice were obtained at 12 weeks of age and acclimated for two weeks prior to dosing (Lines 482-487 of the revised manuscript)

Comment #6

In the description of the phase 1 study design and PK analysis in the methods section, on lines 585-586, it states “Serial blood samples were collected for PK analysis at pre-specified intervals up to 696 hours after the final dose.” The “pre-specified intervals” should be described in the text.

Answer #6

As suggested by the reviewer, the **Methods** section has been revised (Lines 585–587 of the revised manuscript) to specify the exact PK sampling timepoints

Comment #7.

In the description of the statistical analysis in the methods section (line 592), it states “one-way ANOVA with Tukey’s post hoc test was used for multi-group comparisons...” Please specify whether all pairs of groups were tested using Tukey’s post hoc test, or if only a subset of tests of interest were assessed (and if so, which subset of tests were assessed). This will be helpful for interpreting whether lack of an indication of statistical significance in the results section and figures indicates a non-statistically significant test result or simply that the comparison was not tested. In addition, the figure legends

throughout the manuscript state that statistical significance was assessed using one-way ANOVA, but some of the figure legends do not state the use of Tukey's post hoc test; the use of Tukey's post hoc test should be indicated in the figure legends where applicable.

Answer #7

As suggested by the reviewer, all possible pairwise group comparisons were performed using Tukey's post hoc test following one-way ANOVA. Additionally, when multiple treatment groups were compared against a single control, Dunnett's multiple comparisons test was applied. We have revised the **Methods** section (Lines 593–599 of the revised manuscript) to clarify this and updated all relevant figure legends to specify the use of Tukey (all-pairwise) or Dunnett (control-versus-treatment) tests, as appropriate.

Comment # 8

In the figures (e.g., Figure 2), it would be helpful to state the timeframe for cross-sectional results in the figure legend. For example, the legend for Figure Panels 2c and 2d-g state that the results are from end-of-study, but it would be helpful to re-state the timeframe for end-of-study in the legend for easy reference.

Answer #8

As suggested by the reviewer, we have revised the legends for **Figures 2 and 3** to explicitly state the timeframe for all cross-sectional analyses by indicating 'end-of-study (Week 26)

Comment #9.

Figure 6a includes two plots stacked on top of each other. Based on the figure legend, the top plot is on the linear-scale, and the bottom plot presents the same data on the semilogarithmic-scale. It would be helpful to include headings on these two plots to make it clear what is the difference between the two plots.

Answer #9

As suggested by the reviewer, we have added headings to **Figure 6a**: 'Linear scale' (upper panel) and 'Semilogarithmic scale' (lower panel)

Comment #10

Table 1: It is not clear to me what is meant by "treatment-related" adverse events. Please provide a definition for this category of adverse events in the manuscript text and in the description for Table 1.

Answer #10

As suggested by the reviewer, we have added a definition of 'treatment-related' adverse events in the **Methods** section (Lines 590-592 of the revised manuscript). In addition, this definition is now included in the explanatory note below **Table 1** for clarity.

Comment #11

Was Extended Data Table 1 (Thyroid C cell and gastrointestinal hyperplasia in long-term repeated dose toxicity studies in rats and cynomolgus monkeys) referred to anywhere in the manuscript text?

Answer #11

As suggested by the reviewer, **Extended Data Table 1** is referred in the **Results** section (Lines 129–

133 of the revised manuscript) when describing the absence of thyroid C-cell and gastrointestinal hyperplasia in the 26-week repeat-dose toxicity study

Comment #12.

I am a bit confused about screening and enrollment for the phase 1 multiple ascending dose study, based on the data presented in Extended Data Table 2. In particular:

a. It would be helpful to specify in the table heading and/or description that the numbers presented here are for the phase 1 multiple ascending dose study.

Answer #12a

As suggested by the reviewer, we have revised the heading and legend of **Extended Data Table 2** to specify that the numbers refer to the Phase 1 multiple ascending dose study

b. There is no column for the placebo group provided in this table. The relevant information for the placebo group should be added.

Answer #12b

As suggested by the reviewer, placebo group information is incorporated in **Extended Data Table 2**. For clarity, we have revised the table heading to indicate that values are presented as 'PG-102 : Placebo' (e.g., 6:2 indicates 6 subjects received PG-102 and 2 subjects received placebo) across all categories

c. The number of volunteers screened and number of volunteers not enrolled are provided in the table by treatment group. Were participants randomized before enrolling in the study? If so, what was the justification for this?

Answer #12c

As suggested by the reviewer, enrollment was defined as randomization (6:2 per cohort). The 'screened' and 'not enrolled' numbers reflect the pre-randomization process. This has been clarified in the **Methods** section (Lines 570-572 of the revised manuscript), **Extended Data Table 2**, and **Extended Data Figure 10** of the revised manuscript).

d. The numbers of subjects enrolled in this table do not seem to match the sample size provided in the methods section text, or in the other tables related to the phase 1 multiple ascending dose study. This table states that there were 8 people enrolled in each of the 3 PG-102 dose cohorts. However, the methods section text states that there were a total of 24 participants, randomized in a 6:2 ratio for the 3 PG-102 dose cohorts and placebo, implying that there should be 6 participants in each treatment group, including each of the PG-102 dose cohorts. The other tables and figures (e.g., Table 1, Extended Data Table 3, Extended Data Figure 2) also imply a total sample size of 24, with 6 participants in each of the 3 PG-102 dose cohorts and in the placebo group. Please confirm that the sample sizes are correct throughout the manuscript.

Answer # 12d

As suggested by the reviewer, the sample sizes are correct: 24 participants in total, with 8 per cohort randomized in a 6:2 ratio to PG-102 or placebo (18 PG-102, 6 placebo overall). The **Methods** section (Lines 570-576 of the revised manuscript) has been revised to clarify that randomization occurred within each cohort

e. Similar to the above point, in Extended Data Figure 2, the numbers who were screened, not eligible,

and met inclusion/exclusion criteria are consistent with the total numbers across the 3 PG-102 dose cohorts from Extended Data Table 2 (i.e., excluding the placebo group). Should the numbers in Extended Data Figure 2 be revised to include the corresponding numbers from the placebo group, or were the numbers tabulated incorrectly in Extended Data Table 2? Please revise as needed to ensure that the numbers presented in Extended Data Table 2 and Extended Data Figure 2 are both correct.

Answer 12e

As suggested by the reviewer, **Extended Data Figure 2 (Extended Data Figure 10** of the revised manuscript) has been corrected to align with **Extended Data Table 2** of the revised manuscript, and both now include PG-102 and placebo participants with consistent totals

f. Under “Reason for screening failure”, there were a total of 6 participants who failed screening for “other” reasons. Are these the same 6 participants who were excluded from the study due to exceeding the subject limit, according to Extended Data Figure 2? If the reason for screening failure is the same for all 6 participants in the “other” category (i.e., due to exceeding the subject limit), then this reason can be stated in the table, instead of labeling the row as “others”.

Answer 12f

As suggested by the reviewer, the 6 participants listed under ‘other’ were those who exceeded the subject limit. **Extended Data Table 2** and **Extended Data Figure 2 (Extended Data Figure 10** of the revised manuscript) have been revised accordingly for consistency in the revised manuscript

Comment #13

For the phase 1 multiple ascending dose study, based on Extended Data Table 2, there was one participant who dropped out of the study. Did this participant drop out of the study completely (i.e., stopped participating in data collection procedures), or did they just discontinue the randomized medication? Was the participant who dropped out included in any of the analyses, and if so which ones? Please specify this information in the description of the phase 1 multiple ascending dose study in the methods section.

Answer #13

As suggested by the reviewer, one participant in the 15 mg cohort discontinued due to an adverse event unrelated to the investigational product. This participant completed follow-up safety evaluations and was included in the safety analysis set, but excluded from efficacy and PK analyses. This has been clarified in the **Methods** section (Lines 579–583 of the revised manuscript), **Extended Data Figure 2 (Extended Data Figure 10** in the revised manuscript) and **Extended Data Table 2**

Comment #14

Extended Data Table 5: There is an asterisk next to Tmax in the table, but the meaning of the asterisk is not defined anywhere in the table or legend.

Answer #14

As suggested by the reviewer, the unexplained asterisk next to Tmax in **Extended Data Table 5** has been removed, and minor typographical errors have been corrected for consistency in the revised manuscript

Comment #15

Extended Data Figure 1: On line 917 of the figure legend, what is the meaning for the numbers in

parentheses following exendin (9-39) and GLP-2 (3-33)?

Answer #15

The numbers in parentheses indicate the amino acid positions defining the truncated peptide sequences. Exendin (9–39) corresponds to residues 9–39 of exendin-4, acting as a GLP-1 receptor antagonist, and GLP-2 (3–33) corresponds to residues 3–33 of GLP-2, acting as a GLP-2 receptor antagonist. This clarification has been added to the legend of **Extended Data Figure 1** (now **Extended Data Figure 8** of the revised manuscript).

Comment #16

Extended Data Figure 2: In the second box on the right side, the word “consent” is included twice.

Answer #16

As suggested by the reviewer, the duplicated word ‘consent’ in **Extended Data Figure 2** (**Extended Data Figure 10** of the revised manuscript) has been corrected in the revised manuscript

Comment #17.

Extended Data Figure 3:

a. Why was semaglutide excluded from this analysis of the moderate and mild hyperglycemia mouse models (even though it was included in the analysis of the severe hyperglycemia mouse model in Figure 3)? This should be explained in the figure legend and/or in the manuscript text.

Answer #17a

As suggested by the reviewer, we have revised not only the relevant sentence but also the surrounding paragraph (Lines 192–200 of the revised manuscript) to clarify the rationale for excluding semaglutide from the mild and moderate hyperglycemia models.

Semaglutide was not re-tested in these models, as its efficacy had already been confirmed in the severe hyperglycemia model. Instead, the additional studies focused on a direct head-to-head comparison with tirzepatide at an equivalent dose (30 nmol/kg), given its clinical relevance and the attenuated GIPR agonism of tirzepatide in rodents, which warranted focused evaluation of its relative efficacy.

b. Panels d-f (results from the mild hyperglycemia mouse model) appear to be missing.

Answer #17b

As suggested by the reviewer, the data from the mild hyperglycemia model have been added to **Extended Data Figure 4** (new panels d–g), and the corresponding figure legend and Results section have been updated accordingly (Lines 195–197 in the revised manuscript).

c. The dose for tirzepatide appears to be incorrect in the legend indicating the colors and symbols for the different treatment groups (it is listed as 30 nmol/kg, but should be 15 nmol/kg).

Answer #17c

As suggested by the reviewer, we have re-checked the figure legend and confirm that tirzepatide is correctly labeled as 15 nmol/kg in all relevant panels, including the symbol/color legend for **Figure 3**.

No changes were required in the revised manuscript

Comment #18.

Extended Data Figure 4: The procedures for this comparison of PG-105 and PG-102 do not appear to be described in the methods section.

Answer #18

As suggested by the reviewer, the Supplementary Methods section has been updated to include the procedures for the comparison of PG-105 and PG-102 underlying **Extended Data Figure 4 (Extended Data Figure 1)** of the revised manuscript).

Of note, the original panel **d** of **Extended Data Figure 4** (gross photograph of the gallbladder) has been removed to prevent potential visual over-interpretation. Gallbladder size at necropsy is inherently variable—affected by bile filling, handling, and viewing angle—and representative images from a small sample (n=4 per group) cannot reliably represent the group distribution. Moreover, the image could be misread as inconsistent with the quantitative data, which showed a significant increase with PG-105 but no statistically significant change with PG-102. In line with *Nature Communications* guidelines prioritizing quantitative presentation, we now display only the volumetric measurement with individual data points (panel **e** of Extended Data Figure 4, now panel **d** of **Extended Data Figure 1**), which more accurately and reproducibly reflects the dataset.

Reviewer #2

This manuscript presents the preclinical and early clinical development of PG-102, a bispecific GLP-1/GLP-2 receptor agonist, for the treatment of type 2 diabetes (T2D). The authors demonstrate PG-102 in vitro characterization and in vivo efficacy in preclinical models and report favorable tolerability and pharmacokinetics in a Phase 1 multiple ascending dose (MAD) study. The central claim is that PG-102 achieves glycemic control without inducing weight loss, distinguishing it from existing GLP-1-based therapies.

While the manuscript is ambitious and addresses an important clinical need, several key mechanistic claims are insufficiently supported. In particular, the dual pharmacology of PG-102 is not convincingly demonstrated in vivo. Additionally, some experimental designs and interpretations require clarification or further validation.

Major Comments

Comment #1

The manuscript should include a direct in vivo comparison between PG-102, the GLP-1-NTIG Fc, the GLP-2-NTIG Fc, and a loose combination of the GLP-1 and GLP-2 NTIG Fc's to validate the synergistic effects of dual receptor targeting, especially for glucose control and islet protection.

Answer #1

As suggested by the reviewer, we generated three complementary data sets directly comparing PG-102 with GLP-1-NTIG Fc, GLP-2-NTIG Fc, and their loose combination:

1. **Glucose control** (ipGTT; normoglycemic mice, **Figure 5a** of the revised manuscript): At GLP-1R-matched dosing, PG-102 improved glucose clearance relative to GLP-1-NTIG Fc and the loose GLP-1/GLP-2-NTIG Fc combination, while GLP-2-NTIG Fc alone showed minimal effect.

2. **Islet protection under diabetogenic stress** (INS-1 + STZ; **Figure 5b** and **Extended Data Figure 7** of the revised manuscript): PG-102 outperformed each monospecific construct and matched or exceeded the loose combination across GSIS and β -cell gene expression (restoration of *Pdx1* and *Neurod1*; *MafA* showed an upward trend; *TNF α* reduced).

3. **Receptor trafficking** (**Extended Data Figure 9** of the revised manuscript): Confocal imaging demonstrated *cis*-acting co-engagement with delayed internalization and persistent signaling, not reproduced by loose combinations.

To better reflect the reviewer's suggestion, we have substantially revised the manuscript by reorganizing the mechanistic figures, introducing a new Results subsection titled "Bivalent receptor engagement underlying PG-102 efficacy," and restructuring the related text.

Specifically, the **Results** (Lines 201–263) and **Discussion** (Lines 325–365) sections have been thoroughly revised to provide a clearer and more cohesive mechanistic narrative.

Comment #1a

PG-102 exhibits only ~3.4% of native GLP-2R potency. It is unclear whether this level of activity is sufficient to elicit meaningful in vivo effects. A comparison with a GLP-1-only antibody is essential to isolate the contribution of GLP-2R agonism.

Answer #1a

As suggested by the reviewer, direct comparisons with a GLP-1-only construct were conducted (**Figure 5a**, Lines 228-235 of the revised manuscript), along with receptor trafficking studies (**Extended Data Figure 9**, Lines 254-263 of the revised manuscript). Despite exhibiting only ~3.4% of native GLP-2R potency in cAMP assays, PG-102 achieved superior *in vivo* glucose control and demonstrated *cis*-acting GLP-1R/GLP-2R co-engagement with delayed internalization and sustained signaling. These results confirm a meaningful GLP-2R contribution beyond GLP-1R agonism alone.

Comment #1b

While the adipocyte data support dual receptor synergy, similar experiments in islets or *in vivo* are lacking. Without these, the role of GLP-2 on the central pharmacologic action of PG-102 remains speculative.

Answer #1b

As suggested by the reviewer, we now provide β -cell (INS-1 + STZ) and *in vivo* ipGTT datasets showing that PG-102 outperforms both the GLP-1-only construct and the loose GLP-1/GLP-2 combination in glucose control and β -cell endpoints (**Figure 5a-b**, Lines 228-243 of the revised manuscript). In addition, receptor trafficking analyses (**Extended Data Figure 9**, Lines 254-263 of the revised manuscript) further support a direct GLP-2R contribution, establishing that the role of GLP-2 in PG-102's pharmacology is non-speculative.

Comment #2

The manuscript claims enhanced tolerability in humans, but it is unclear what this is relative to. No marketed comparators were included in the Phase 1 study. Please clarify the basis for this claim.

Answer #2

As suggested by the reviewer, the manuscript has been revised to remove any implicit comparative language. The **Title** was shortened, and tolerability is now described descriptively (low incidence of gastrointestinal events, no treatment-related discontinuations) in the **Discussion** (Line 402–407) of the revised manuscript.

Comment #3

The authors should report whether PG-102 activates other class B GPCRs such as GCGR or GIPR. This is particularly important given the structural similarity and potential for off-target effects.

Answer #3

As suggested by the reviewer, we performed the off-target profiling experiments, as described in the revised **Results** section (Lines 123–128) and **Extended Data Figure 2**. PG-102 showed no detectable agonist activity at human GIPR or GCGR, confirming selectivity for GLP-1R and GLP-2R.

Comment #4

The manuscript uses tirzepatide as a comparator in rodents, but its GIPR activity is limited in this species. This limitation should be acknowledged and discussed.

Answer #4

As suggested by the reviewer, the limitation of tirzepatide's attenuated GIPR agonism in rodents has been acknowledged in the **Results** (Lines 192–200 and 206–208 of the revised manuscript), noting that these data serve as comparative references rather than predictors of clinical efficacy.

Comment #5

The reported 40% spontaneous weight loss in vehicle-treated db/db mice is unusually high. The authors should provide citations or additional data to support this observation. The supplementary data show only ~7% weight loss in a milder glucose intolerance model, which raises concerns about reproducibility.

Answer #5

As suggested by the reviewer, we have cited supporting evidence in the **Discussion** (Line 372) showing that the ~40% spontaneous weight loss observed in the severe hyperglycemia model is consistent with prior reports in advanced-stage *db/db* mice (Choi et al., *Lab Anim Res.* 2015; 31:1–6). The apparent difference from the milder model (~7% loss) is addressed in the revised **Discussion** (Lines 372–375), which clarifies that it reflects baseline disease severity and independent study conditions.

In addition, to more coherently address the mechanisms underlying PG-102's distinctive profile—achieving normoglycemia while substantially preserving body weight across both severe and moderate hyperglycemia models—we have refined the entire **Discussion** paragraph (Lines 366–401) for logical flow and mechanistic clarity.

Comment #6

The authors mention reduced CNS exposure as a potential mechanism for improved tolerability. Have they performed ICV injections or biodistribution studies to support this?

Answer #6

As suggested by the reviewer, we have further clarified the rationale regarding potential CNS exposure in relation to tolerability.

While intracerebroventricular (ICV) injections or formal CNS biodistribution studies have not yet been performed for the current PG-102 molecule, exploratory imaging with a precursor form of PG-102 (PG-105) demonstrated negligible brain signal and preferential localization to the small intestine, whereas dulaglutide exhibited measurable brain accumulation and predominant pancreatic uptake (**Figure 2** below).

Because these findings were exploratory and based on a precursor molecule, they were not included as main data.

Instead, the conceptual implication—that limited CNS exposure, likely related to the large-molecule nature of Fc-based GLP-1 agonists, may contribute to PG-102's favorable tolerability profile—has been concisely reflected in the **Discussion** section (Lines 415–421 of the revised manuscript) in reference to a previous report supporting reduced CNS penetrance of Fc-fused incretin analogs.

Figure 2. Exploratory biodistribution of PG-105 (a precursor form of PG-102) in Balb/c nude mice. Male BALB/c nude mice (5 weeks old) were administered a single subcutaneous injection of Fc protein, PG-105, or dulaglutide. Whole-body fluorescence imaging was performed at 3, 10, 24, 72, and 168 hours after dosing, followed by ex vivo organ imaging at 10 hours, when blood concentrations were highest. Relative distribution was quantified in major organs including the brain, pancreas, heart, stomach, liver, kidney, small intestine (duodenum, jejunum, ileum), and large intestine. Data are presented as mean \pm SD.

Comment #7

The claim that PG-102 enhances insulin secretion over semaglutide and tirzepatide is not statistically significant (line 215). Similarly, *MafA* expression is not significantly upregulated. These points should be revised or qualified.

Answer #7

As suggested by the reviewer, we repeated the GSIS and *MafA* analyses under the STZ-induced diabetogenic condition with additional biological replicates. The **Results** section has been revised to qualify these findings accordingly (Lines 215–218 of the revised manuscript), indicating that insulin secretion (GSIS) was statistically significant, whereas *MafA* expression showed a non-significant upward trend. Exact n values and statistical details are now provided in the corresponding figure legends.

Comment #8

The manuscript claims anti-inflammatory effects of PG-102 (line 359), but no supporting data are shown. These claims should be either substantiated or removed.

Answer #8

As suggested by the reviewer, we have substantiated the anti-inflammatory claim with new *in vivo* and *in vitro* data. The revised manuscript now includes CD45⁺ immunostaining of peri-pancreatic adipose tissue and TNF α expression analyses under β -cell stress, both supporting anti-inflammatory activity of PG-102 (**Extended Data Figures 3 and 7d**). To integrate these new findings coherently, we have

substantially revised the relevant Results (Lines 183–191) and Discussion (Lines 342–352) sections to strengthen the mechanistic narrative regarding PG-102’s anti-inflammatory actions.

Comment #9

Plot the AUC for glucose and drug exposure (Figure 6a vs Figure 6b) to show the relationship between exposure and response.

Answer #9

As suggested by the reviewer, an exploratory exposure–response analysis was added, plotting PK exposure (AUC_{last}) against OGTT glucose AUC_{0-2h} . The corresponding results are described in the **Results** (Lines 303–304 of the revised manuscript) and presented in **Extended Data Figure 11**.

Minor Comments

Comment #1. Line 84: Explicitly state the goal of achieving glycemic control without weight loss.

Answer #1

As suggested by the reviewer, we have revised the **Introduction** to explicitly state the study objective of achieving glycemic control without weight loss (Line 56–61 of the revised manuscript)

Comment #2. Line 109: Clarify that no marketed comparators were included in the Phase 1 study.

Answer #2

As suggested by the reviewer, we have revised the manuscript to clarify that no marketed comparators were included in the Phase 1 study and removed any phrasing that could imply otherwise.

Comment #3. Line 218: Cite El, Douros et al. Nature Metabolism 2023 to support the role of GIPR in the pharmacology of tirzapatide.

Answer #3.

As suggested by the reviewer, we have added a citation to El Douros et al., Nature Metabolism (2023) at Line 192 of the revised manuscript to support the role of GIPR in the pharmacology of tirzapatide.

Comment #4. Line 324: Replace “potent” with “effective” to avoid overstatement.

Answer #4

As suggested by the reviewer, we have replaced ‘potent’ with ‘effective’ at Line 313 of the revised manuscript to ensure balanced wording.

Comment #5. Lines 336–337: Provide data or citations for feline case studies.

Answer #5

The exploratory statement regarding naturally diabetic feline patients (Lines 336–337 of the original submission) has been removed from the revised manuscript because of the following reasons.

The observations originated from pilot cases conducted independently at a collaborating veterinary hospital in Korea, in which naturally diabetic cats were treated at the discretion of the attending veterinarian with verbal consent obtained from the pet owners. At the time of the initial manuscript preparation, our understanding was that Korea did not yet have clearly defined regulatory pathways for companion-animal research, and such observational cases could be described anecdotally. However, upon further review during the revision process, we recognized that international standards—particularly those aligned with U.S. and European guidelines—require companion-animal studies to undergo formal ethical review and to obtain documented owner consent analogous to human clinical research.

In compliance with these ethical principles, all feline-related content has been removed from the revised manuscript. For transparency, we provide here a concise descriptive summary and illustrative image (Figure 3) only within this reviewer response. These pilot observations were not designed or statistically powered as controlled experiments and are therefore not presented as validated preclinical or clinical data. The revised manuscript now focuses exclusively on validated preclinical models and the controlled human Phase 1 study, ensuring both scientific rigor and adherence to ethical research standards.

Figure 3. Exploratory glucose-lowering effects of PG-102 in naturally diabetic cats. Two diabetic cats (Cat A: domestic shorthair, 16.8 years, obese; Cat B: Russian Blue, 1.9 years, underweight) were treated with PG-102 (0.3 mg/kg, subcutaneous, once weekly for 4 weeks). Fasting blood glucose, body weight, and serum cholesterol were monitored during follow-up.

Comment #6. Lines 348–354: Add citations to support mechanistic claims.

Answer #6

As suggested by the reviewer, we have added a supporting reference at Line 340 of the revised manuscript to substantiate the mechanistic claims

Comment #7. Line 358: Specify which other cell types co-express GLP-1R and GLP-2R.

Answer #7

As suggested by the reviewer, we have specified in the revised manuscript that GLP-1R and GLP-2R are co-expressed in enteroendocrine L cells, enteric neurons, intestinal epithelial subsets, pancreatic islets, adipocytes and subsets of vagal afferent neurons (Lines 359–361 of the revised manuscript)

Comment #8. Line 367–370: Reframe speculative mechanisms as hypotheses rather than conclusions.

Answer #8

As suggested by the reviewer, we have revised the **Discussion** section to frame the mechanistic interpretations of PG-102 as hypotheses supported by available evidence, rather than as definitive conclusions. In doing so, we extensively restructured the **Discussion** (Lines 325–365) to more clearly delineate evidence-based findings from conceptual interpretations.

Comment #9. Line 388: Show the data referenced.

Answer #9

As suggested by the reviewer, we have added the corresponding data as **Extended Data Figure 12** and incorporated a brief description into the **Discussion** section (Lines 388–392 of the revised manuscript).

Comment #10. Line 431: If available, include biodistribution data for PG-102 and dulaglutide.

Answer #10

Exploratory biodistribution results comparing a precursor form of PG-102 (PG-105) and dulaglutide have been included in response to Major Comment #6 (**Figure 2**) and conceptually acknowledged in the revised **Discussion** (Line 415–421 of the revised manuscript).

Reviewer #3

Strengths

Innovative dual-agonist design with translational potential

The study introduces a well-engineered GLP-1/GLP-2 dual-agonist supported by a mechanistically sound rationale. The combination of GLP-1R-mediated glycaemic control and GLP-2R-driven intestinal and systemic homeostasis offers a potential therapeutic niche in advanced type 2 diabetes, cachexia, or malabsorption.

Comprehensive early-phase study design

The manuscript captures a broad spectrum of tolerability, exposure, and PD signals in healthy adults. The inclusion of multiple ascending dose arms and detailed PK/PD endpoints (including DEXA, OGTT, HbA1c, FMI/LMI) supports an informative pharmacological characterization. Albeit, the vast majority of these variables are not presented in the current manuscript.

Preclinical comparator data strengthen translational appeal

Head-to-head comparisons with semaglutide and tirzepatide in db/db mice highlight superior glucose-lowering efficacy independent of body weight, suggest a favorable pharmacodynamic dissociation.

Weakness

Comment #1

No formal statistical hypothesis or power justification provided

Although Phase 1 trials are typically exploratory, the manuscript would benefit from explicit clarification regarding sample size rationale and statistical assumptions for key PD endpoints (e.g., body weight change, HbA1c).

Answer #1

As suggested by the reviewer, this Phase 1 MAD study was designed to evaluate safety, tolerability, and PK; no formal hypothesis testing or power calculation was performed. The cohort size (n=8 per cohort; total n=24) follows standard early-phase practice. PD and exploratory biomarkers were analyzed descriptively, as clarified in the **Methods** section (Line 562-565 of the revised manuscript).

Comment #2

Limited discussion of GLP-2R-specific effects in humans (efficacy and safety)

While the dual-agonist concept is central, the manuscript provides little evidence or speculation about the contribution of GLP-2R agonism per se. For instance, GI permeability markers, gut hormones, or nutrient absorption were not assessed, weakening the mechanistic interpretation of the GLP-2R arm. Especially, the safety concerns related to long-term GLP-2 RAs in humans linked to intestinal neoplasms needs to be addressed.

Answer #2

As suggested by the reviewer, we have clarified in the **Discussion** (Lines 431–437 of the revised manuscript) that GLP-2R-specific endpoints such as intestinal permeability, nutrient absorption, or gut hormone secretion were not directly assessed in this phase 1 study. Regarding long-term safety, we have added a statement citing recent real-world data showing no increase in intestinal neoplasms with long-term teduglutide use (Abu Tair et al., Intestine Failure Reports, 2025) and noting that 26-week repeat-

dose toxicity studies of PG-102 in rodents and primates showed not only an absence of intestinal hyperplasia but also no evidence of thyroid C-cell hyperplasia commonly associated with GLP-1RAs (**Extended Data Table 1** and Lines 437–444 of the revised manuscript).

Comment #3

PD effects in humans appear modest

Only glycaemic effects of PG-102 treatment are presented in this manuscript which is insufficient to thoroughly assess the effects of the compound. Body weight, body composition and hsCRP data needs to be presented as well. But most importantly, the effect of PG-102 appears fairly modest compared to placebo treatment. This may reflect high baseline insulin sensitivity, insufficient duration, or limited GLP-1R potency at early dose levels. The glucometabolic effect of GLP-2 receptor activation in humans remain controversial.

Answer #3

As suggested by the reviewer, we have added exploratory data on body weight, body composition, and inflammatory biomarkers (hsCRP), along with glycemic endpoints (HbA1c and FPG), to **Extended Data Tables 6 and 7**. As expected in healthy volunteers with high baseline insulin sensitivity and a short treatment duration, changes in these parameters were modest (**Results**, Lines 305–306 of the revised manuscript). Importantly, PG-102 consistently reduced glucose AUC during OGTT across active cohorts, demonstrating clear biological activity even at early dose levels (**Results**, Lines 298-304 of the revised manuscript). Our findings suggest that GLP-2R activation may contribute meaningfully to glucometabolic regulation, a hypothesis that will be further examined in Phase 2 studies involving diabetic patient populations.

Overall level of enthusiasm

This is a well-conceived and well-executed early-phase translational study introducing a novel dual GLP-1/GLP-2 receptor agonist delivering exciting glycaemic results independent of change in body weight in rodents. Although the human PD effects are modest and the absence supporting data (insulin, Matsuda-index, change in body weight, body composition and inflammatory cytokines) are disappointing. The preclinical results (particularly in the db/db model) are compelling.

The manuscript will be of interest to the field of incretin biology and metabolic pharmacology, but it is very heavy on the preclinical which is a disappointment. I recommend publication pending revision and the addition of more human data.

Additional comments

Below are some suggestions aiming to improve the manuscript.

- In general

Comment #1. Line 319: In T2D, metformin is the first-line treatment for hyperglycemia. Please correct the sentence

Answer #1

As suggested by the reviewer, we have revised the text to clarify that metformin is the standard first-line therapy for T2D, whereas insulin is recommended for patients with severe hyperglycemia (HbA1c >10%) in line with ADA/EASD guidelines (Lines 308-310 of the revised manuscript).

Comment #2. Line 325: Please provide insight into how many participants in the STEP-2 or STEP-6 and SURPASS trials obtain ‘treat-to-target’ success with an HbA1c below 5.7%

Answer #2

As suggested by the reviewer, we have revised the **Discussion** (Line 313-316 of the revised manuscript) to clarify that data on ‘treat-to-target’ success with HbA1c <5.7% are available from the SURPASS trials, where fewer than 50% of participants achieved normoglycemia even at the highest tirzepatide dose. The STEP trials primarily evaluated obesity populations rather than T2D cohorts; therefore, they do not directly report HbA1c-based remission outcomes.

Comment #3. Line: 348-353: Please provide references supporting these claims.

Answer #3

As suggested by the reviewer, we have added supporting references at Line 340 of the revised manuscript to substantiate these mechanistic claims.

Comment #4. Line 359: The author has not provided data supporting this claim of anti-inflammatory effects of PG-102

Answer #4

As suggested by the reviewer, we have added new experimental data to substantiate the anti-inflammatory effects of PG-102. These include reduced CD45⁺ immune cell infiltration in pancreatic adipose tissue and suppression of TNF α induction under β -cell stress (**Extended Data Figures 3 and 7d** of the revised manuscript). In line with these additions, both the **Results** (Lines 183–191 and 240-243) and **Discussion** (Lines 342–352) have been revised to more clearly articulate the mechanistic link between PG-102’s dual-receptor engagement and its anti-inflammatory profile.

Comment #5. Line 370: Please update this section with a remark alluding to the anti-inflammatory effects observed in the present study. If, please inform the reader, that the anti-inflammatory effects of PG-102 otherwise are purely speculative

Answer #5

As suggested by the reviewer, this point was addressed in our response to Comment #4. The revised **Discussion** section (Lines 342–352) now incorporates the newly added anti-inflammatory findings, thereby moving beyond speculative interpretation.

Comment #6. Line 371-374: Please provide reference

Answer #6

As suggested by the reviewer, we have added the appropriate reference to support this statement (Line 368 of the revised manuscript).

Comment #7. Line 590: Clarify whether any correction for multiple testing was applied across PD endpoints.

Answer #7

As suggested by the reviewer, we have clarified in the **Methods** section (Lines 600–601 of the revised manuscript) that PD assessments were exploratory and summarized descriptively. For endpoints with statistical testing, appropriate post hoc tests were applied as described, but no formal correction across multiple PD endpoints was performed.

Comment #8. Line 596: Please add supporting information to the sample size calculation

Answer #8

As suggested by the reviewer, we have clarified the basis of the sample size determination in the Statistical Analysis section of the **Methods** section (Lines 487–489 of the revised manuscript). Specifically, we now state that sample sizes were guided by prior experience with the *db/db* mouse model and designed to provide at least 80% power at $p < 0.05$ for detecting biologically meaningful differences in metabolic outcomes.

Comment #9. Line 809 (Fig. 6): Please provide more details with regards to exact N per timepoint if sample size varies

Answer #9

As suggested by the reviewer, all participants included in the PK analysis completed every planned sampling timepoint without missing data; thus, n was identical across timepoints (15 mg: n = 5; 30 mg: n = 6; 30/60 mg: n = 6; placebo: n = 6). The legend of **Figure 6** has been revised to clarify this.

Comment #10. Line 924 (Extended Data Fig. 2.): please add reason for dropout.

Answer #10

As suggested by the reviewer, we have clarified that in the 15 mg cohort, one participant discontinued due to an adverse event unrelated to the investigational product. This information has been added to the **Methods** (Line 579–583 of the revised manuscript), **Extended Data Figure 2** (**Extended Data Figure 10** of the revised manuscript), and **Extended Data Table 2**

Reviewer #1

This is a revised manuscript that evaluates PG-102, which targets both GLP-1 and GLP-2 receptors to treat type 2 diabetes, based on in vitro studies, mouse studies comparing to semaglutide and tirzepatide, and a phase 1 multiple ascending dose study in humans. The revised manuscript is much clearer and improved, and in my opinion it is nearly ready for publication. There are a few minor remaining issues that should be addressed, which are described below:

Comment #1

In the Results, “Phase 1 MAD trial of PG-102: safety, PK and PD evaluation” section, on line 273: I would suggest clarifying that the 6:2 randomization ratio is for PG-102:placebo. For example, the sentence could be revised as “...randomized in a 6:2 ratio to receive PG-102 or placebo within three dose cohorts...”:

Answer #1

As suggested by the reviewer, the sentence has been revised to clearly specify that the 6:2 randomization ratio corresponds to PG-102 versus placebo allocation within each cohort (Line 281 of the revised manuscript).

Comment #2

In the Results, “Phase 1 MAD trial of PG-102: safety, PK and PD evaluation” section, on lines 283-284: Since the AEs can be classified as treatment-related, not the participants, I suggest re-wording the last part of this sentence as “...with AEs considered to be treatment-related in 18 participants (75.0%; Table 1).

Answer #2

As suggested by the reviewer, we revised the wording to clarify that treatment-relatedness applies to AEs rather than participants, and updated the sentence accordingly (Lines 293 of the revised manuscript).

Comment #3

In the Methods, “Phase 1 study design; PK analysis, safety and PD evaluation” section, lines 560-561: As this sentence is currently written, it is a bit confusing since it refers to the three cohorts from the MAD study without providing information defining the cohorts. Most of the information from this sentence is clearly included later in this section on lines 570 – 572, and so the sentence from lines 560 – 561 could be deleted.

Answer #3

As suggested by the reviewer, the redundant sentence (Lines 560-561 of the original manuscript) referring to cohort composition was removed to improve clarity, as the cohorts are defined later in the Methods section (Lines 632–634 of the revised manuscript).

Comment #4

Please define all abbreviations from the figures and tables in the corresponding figure legend or table

footnote, including SEM, SD, PK, PD, MAD, AUC.

Answer #4

As suggested by the reviewer, all abbreviations (SEM, SD, PK, PD, MAD, AUC) are now fully defined in the relevant figure legends and table footnotes.

Comment #5.

The last sentence in the legend for Figure 6 states “All randomized participants completed every planned PK and PD sampling time point.” However, based on the Methods (lines 579-580) and the sample sizes listed in the figure legend, it seems that this statement is not quite accurate, since 1 randomized participant in the 15 mg group who discontinued treatment was excluded from the PK/PD analyses. Instead, this sentence could be re-written as “All participants included in the analyses completed every planned PK and PD sampling time point.”

Answer #5

As suggested by the reviewer, the Figure 6 legend has been revised to specify that all participants included in the PK and PD analyses completed all planned sampling time points (Lines 957-958 of the revised manuscript)

Comment #6

In Table 1, the number of participants with discontinuation due to AEs for the 15 mg PG-102 is listed as 0. Is this correct? Based on the Methods (lines 579-580), it seems that the 15 mg group had 1 discontinuation due to AEs (but there were 0 discontinuations due to treatment-related AEs).

Answer #6

As suggested by the reviewer, we have revised the table (**Table 2** of the revised manuscript) to specify “discontinuation due to treatment-related adverse events” and added a clarifying note in the table footnote.

Comment #7

In the Supplementary Methods, “In vivo study” section, lines 35-36: This sentence states that the mice were “allocated by baseline glucose, HbA1c, and body weight into three groups.” Does this mean that treatment group allocation was designed to achieve balance of baseline glucose, HbA1c, and body weight across groups (e.g., by “matching” mice based on these factors)? If so, I suggest that the following re-wording of this sentence would more clearly explain how this allocation was done: “...allocated into three groups (vehicle, tirzepatide, PG-102; n=6 per group), ensuring between-group balance on baseline glucose, HbA1c, and body weight.”

Answer #7

As suggested by the reviewer, we have revised the wording in the Supplementary Methods (Lines 34-37) to clarify that mice were allocated into treatment groups in a manner that ensured between-group balance with respect to baseline glucose, HbA1c, and body weight.

Comment # 8

In the Clinical Supplementary Results, “Exploratory exposure-response relationship between PK exposure and OGTT glucose excursion” section: Be careful not to overstate the relationship identified between PK exposure and OGTT glucose excursion. The R^2 , Pearson r , and Spearman ρ are all small in magnitude, and so I would not have much confidence about whether the negative relationship identified represents a real biological relationship vs. is due to random error.

Answer #8

As suggested by the reviewer, we have revised Lines 167-174 of the Clinical Supplementary Results to avoid overinterpretation of the exploratory exposure–response analysis by explicitly noting the lack of statistical significance, and the limited sample size.

Comment #9.

In Extended Data Table 5: Note that the notation $AUC_{\{0-\text{inf}\}}$ is used in the table, but the notation $AUC_{\{0-\infty\}}$ is used in the table description. Please revise so that the notation is consistent between the table and the description.

Answer #9

As suggested by the reviewer, we have revised the table legend of Extended Data Table 5 (Extended Data Table 4 of the revised manuscript) to ensure consistent notation of AUC. The legend has been updated to use “ $AUC_{0-\text{inf}}$,” matching the notation used in the table.

Reviewer #4

The revised manuscript comprehensively addresses the original comments from Reviewer 3 and strengthens the scientific rigor of the work. The integration of preclinical and clinical data supporting PG-102 as a bispecific GLP-1/GLP-2 receptor agonist for advanced type 2 diabetes (T2D) is compelling, particularly its ability to decouple glycemic control from weight loss. However, 1 key area requires further refinement to enhance mechanistic clarity, generalizability, and translational relevance. Below are specific suggestions for revision:

Comment #1

Mechanistic Link Between Low GLP-2R Potency and In Vivo Efficacy

PG-102 exhibits only ~3.4% of native GLP-2R potency in vitro (Fig. 1c) but delivers meaningful in vivo effects (e.g., enhanced peripheral glucose uptake, β -cell protection). While the manuscript attributes this discrepancy to “cis-co-engagement” of GLP-1R/GLP-2R, direct evidence linking this receptor trafficking phenotype to downstream GLP-2R-mediated signaling in human cells/tissues is lacking.

Add a discussion section explicitly addressing how cis-co-engagement compensates for low GLP-2R potency, citing relevant literature on receptor trafficking and signaling crosstalk.

Answer #1

As suggested by the reviewer, we have added a dedicated Discussion paragraph addressing how bivalent cis co-engagement of GLP-1R and GLP-2R could reconcile PG-102's attenuated intrinsic GLP-2R potency (~3.4% of native GLP-2 in vitro; Fig. 1c) with its robust in vivo efficacy. We also acknowledge that direct evidence linking the phenotype to GLP-2R-mediated downstream signaling in human tissues is currently lacking and identify this as an important area for future validation. These revisions have been incorporated into the revised manuscript in the Discussion (Lines 363–387).